# Effects of Montmorillonite and Gentamicin Addition on the Properties of Electrospun Polycaprolactone Fibers

**DOI:** 10.3390/ma14226905

**Published:** 2021-11-16

**Authors:** Ewa Stodolak-Zych, Roksana Kurpanik, Ewa Dzierzkowska, Marcin Gajek, Łukasz Zych, Karol Gryń, Alicja Rapacz-Kmita

**Affiliations:** 1Department of Biomaterials and Composites, AGH University of Science and Technology, 30-059 Krakow, Poland; kurpanik@agh.edu.pl (R.K.); dzierzkowska@agh.edu.pl (E.D.); kgryn@agh.edu.pl (K.G.); 2Department of Ceramics and Refractories, AGH University of Science and Technology, 30-059 Krakow, Poland; mgajek@agh.edu.pl (M.G.); lzych@agh.edu.pl (Ł.Z.)

**Keywords:** nanobiocomposites, montmorillonite, electrospinning, nanocomposites, antibacterial properties, mechanical properties, fibrous membrane

## Abstract

Electrospinning was used to obtain multifunctional fibrous composite materials with a matrix of poly-ɛ-caprolactone (PCL) and 2 wt.% addition of a nanofiller: montmorillonite (MMT), montmorillonite intercalated with gentamicin sulphate (MMTG) or gentamicin sulphate (G). In the first stage, the aluminosilicate gallery was modified by introducing gentamicin sulfate into it, and the effectiveness of the intercalation process was confirmed on the basis of changes in the clay particle size from 0.5 µm (for MMT) to 0.8 µm (for MMTG), an increase in the interplanar distance *d_001_* from 12.3 Å (for MMT) to 13.9 Å (for MMTG) and altered clay grain morphology. In the second part of the experiment, the electrospinning process was carried out in which the polymer nonwovens with and without the modifier were prepared directly from dichloromethane (DCM) and N,N-dimethylformamide (DMF). The nanocomposite fibrous membranes containing montmorillonite were prepared from the same polymer solution but after homogenization with the modifier (13 wt.%). The degree of dispersion of the modifier was evaluated by average microarray analysis from observed area (EDS), which was also used to determine the intercalation of montmorillonite with gentamicin sulfate. An increase in the size of the fibers was found for the materials with the presence of the modifier, with the largest diameters measured for PCL_MMT (625 nm), and the smaller ones for PCL_MMTG (578 nm) and PCL_G (512 nm). The dispersion of MMT and MMTG in the PCL fibers was also confirmed by indirect studies such as change in mechanical properties of the nonwovens membrane, where the neat PCL nonwoven was used as a reference material. The addition of the modifier reduced the contact angle of PCL nonwovens (from 120° for PCL to 96° for PCL_G and 98° for PCL_MMTG). An approximately 10% increase in tensile strength of the nonwoven fabric with the addition of MMT compared to the neat PCL nonwoven fabric was also observed. The results of microbiological tests showed antibacterial activity of all obtained materials; however, the inhibition zones were the highest for the materials containing gentamicin sulphate, and the release time of the active substance was significantly extended for the materials with the addition of montmorillonite containing the antibiotic. The results clearly show that the electrospinning technique can be effectively used to obtain nanobiocomposite fibers with the addition of nonintercalated and intercalated montmorillonite with improved strength and increased stiffness compared to materials made only of the polymer fibers, provided that a high filler dispersion in the spinning solution is obtained.

## 1. Introduction

Nanobiocomposite materials are materials in which one of the components, i.e., the matrix (matrix) or the filler (modifying phase), is of natural origin, such as, montmorillonite. These materials have gained increasing popularity thanks to safe decomposition products: clay, which is recycled into the environment intact, and a biodegradable synthetic polymer, whose decomposition products, lactide, glycol or caprolactone units, are utilized by microorganisms (bacteria, fungi) to carbon dioxide and water [1,2]. The addition of natural clay to the degradable polymer matrix increases the possible applications of such a nanobiocomposite material but also has a positive effect on its mechanical properties [3], barrier properties appear [4], thermal properties change [5], flammability decreases [6] and water absorption increases [7]. The layered aluminosilicate nanofiller itself may also be a potential “source” of modification. The skillful intercalation of the interlayer spaces of the clay enables it to be given the role of a carrier, thanks to which the poly-ε-caprolactone (PCL)/montmorillonite (MMT) nanobiocomposite acquires new properties, e.g., antibacterial properties extended in a controlled time. Such material solutions can be used in the food packaging industry or in biomedical engineering (dressings, scaffolds and drug carriers) [8,9,10].

The possibility of applying nanocomposites with the addition of MMT, in this respect, results from their better mechanical and thermal properties [11,12,13,14,15]. In addition, based on the processing capabilities of the matrix, it is possible to form such plastics using injection, extrusion or casting [11,12,13]. Unfortunately, in all solutions, there is a problem related to the homogeneous distribution of nanofillers. If, for the purposes of forming a nanocomposite, the methods of processing thermoplastics (e.g., PLA) are used, preintercalated clay is usually added as a modifier, and this preparation of the filler improves the dispersion during the homogenization process. A frequent additive facilitating the compatibility of clay is monomers, e.g., PCL monomer [14]. On the basis of the obtained results of research on mechanical properties, it was found that the addition of clay and PCL compatibilizer significantly increases the strength and stiffness of PLA/PCL nanocomposites with MMT. In addition, the storage modulus of the material increases, which can be used to predict the behavior of the material during operation (in tension) and under the influence of temperature. In turn, the casting method is based on dissolving the polymer in an organic solvent and improving the wettability of the clay nanoparticles during the initial process preceding the dispersion [15]. Such systems use modification of the clay with ammonium salts often based on commercially available products, e.g., Cloisite Na, Cloisite 30B and Cloisite 20A. Improving the filler–polymer matrix interaction often results in structural changes (increasing the degree of crystallinity and thermal stability) and changes in mechanical properties (strength and Young’s modulus) [16]. On the other hand, a number of solutions in the field of nanocomposite materials based on polyesters are intended to give them an antibacterial function and/or induce a controlled process of material degradation. In this respect, additives are mainly copper, zinc or titanium oxides or nanoparticles of zinc and titanium [17,18,19]. The proposed solution, i.e., fibrous nanocomposites with MMT intercalated with gentamicin, can play the role of a multifunctional nanocomposite material, and therefore antibacterial and at the same time degradable under an environmental biodegradation process.

Another area of application of nanocomposite systems based on aliphatic polyesters and nanoclays is biomedical engineering, and the popularity of solutions in this area, as in the case of packaging, results from the multitude of processing methods for polymer matrices and the possibility of modifying the filler itself, e.g., with antibiotics and monomers [20,21,22]. Our previous work has shown that the injection method can produce nanocomposites of PLA and modified clay, in which the amount of filler determines the durability of the material in highly hydrated conditions (the more filler, the greater the decrease in molecular weight and mechanical properties of the nanocomposite) [23]. On the other hand, cast PLA/MMT materials modified with gentamicin showed strong antibacterial activity against Gram-negative bacteria such as *E. coli* [24]. The use of the salt washing method allows, in turn, it to obtain porous substrates, the pore size of which depends on the size of the porogen (salt) grains. The introduction of unmodified MMT and modified MMT led to the substrate having gradient properties due to the sedimentation processes that take place during drying [25].

An innovative solution may be the use of antibacterial particles as carriers-layered aluminosilicates, which are cheap and common, and the method of introducing the active substance between the clay packages is relatively easy and repeatable [26,27]. The possibility of modifying layered aluminosilicates offers the potential for a controlled and long-lasting antiseptic effect. One example of such a solution is the introduction of silver ions into the interlayer of clay, which has an antibacterial effect against *E. coli* bacteria [28]. Other examples in the literature show that the MMT nanoadditive can be effectively intercalated with active substances such as: gentamicin [28], poloxamer [10], chlorhexidine acetate [26,29] and nisin [30]. The authors of these solutions indicate the application potential of such antibacterial nanoadditives in the textile industry (using a cotton matrix) [26] or the consumer goods industry, e.g., melamine dishes, hospital rails or elevator buttons [10].

The weakness of all nanofillers is their difficulty to disperse in the polymer matrix and the necessity to use many procedures facilitating homogenization and preventing secondary agglomeration during the process of forming a nanocomposite material. Hence, extensive work is devoted to the methodology of obtaining homogeneous materials, including biodegradable nanocomposites based on polymers. For example, Pantoustier et al. [31] used the method of in situ polymerization to obtain nanocomposites based on PCL. The comparison of the properties of nanocomposites obtained with pure MMT and MMT modified with aminodecanoic acid shows how important is the homogeneous distribution of the filler in the matrix in terms of mechanical and functional properties. In turn, Di et al. [32] described the preparation of nanocomposites from PCL with the use of layered aluminosilicates using a twin-screw extruder and various organic modifications of aluminosilicates improving the compatibility of the filler with the matrix and thus achieved a different degree of nanoclay dispersion in the matrix. Another method of improving the homogeneity and compatibility between the degradable matrix and MMT was demonstrated by Paul et al. [33], who applied melt intercalation with MMT modified with bis-(2-hydroxyethyl) methyl (hydrogenated tallowalkyl) ammonium cations. The solvent method is rarely used for this purpose, since its last stage-drying often leads to uncontrolled changes in the properties of the filler and its secondary agglomeration [34,35]. In terms of these reports, it seems that not only the new properties but also the method of forming the nanocomposite material are a technological challenge.

Taking into account the possible applications of nanobiocomposite materials of bactericidal nature, electrospinning seems to be a good solution as one of the methods of forming ultrathin fibers with a diameter from a few micrometers to nanometers [36,37]. In this method, the potential difference between the nozzle (which is charged with a high voltage) and the collector (which is grounded) causes the extraction of a polymer stream, which is stretched into a fibrous form in an electrostatic field with a sufficiently high DC voltage [36]. If, at the stage of preparation of the spinning solution, a nanoadditive, e.g., in the form of particles, is introduced into the polymer solution, then the fibers deposited on the collector are nanocomposite fibers. Such materials are characterized by a high surface-to-volume ratio, which leads to the improvement of many properties, from chemical to physicochemical and mechanical, and simultaneously allows for a relatively high homogeneity of the material in terms of the presence of the additive in the matrix [37,38,39]. Membranes composed of nanofibers are characterized by high surface energy and often also high hydrophobicity due to the synergistic effect of ultrathin fibers and the nature of the polymer itself [40,41]. By introducing nanoparticles into the matrix of the fiber, the wettability of the surface of fibrous materials can be controlled, as is the case in PCL/HAp or PCL/TCP systems [42].

Polycaprolactone as a base material for nanocomposite production is a solution already used in the literature. Aliphatic polyester with a relatively long shelf life (even up to 2 years) but proven biocompatibility is the basis of materials for tissue engineering and regenerative medicine. The ease of processing and the low melting point are often used in the development of various PCL-based material forming techniques [43,44,45].

The analysis of the literature clearly shows the potential applications of nanobiocomposite materials, which is why the aim of the study was to produce fibrous membranes by electrospinning, consisting of PCL and layered aluminosilicate-montmorillonite (MMT), which was previously intercalated with gentamicin sulfate (G). In the first part, the effectiveness of MMT intercalation with gentamicin sulfate was characterized, showing the optimal conditions for obtaining MMTG combination. The effectiveness of intercalation was confirmed by the X-ray method (XRD) and dynamic light scattering (DLS). In the further part of the work, a number of membrane materials were prepared: neat PCL without additives as a reference, PCL with the addition of montmorillonite (PCL_MMT), PCL with the addition of gentamicin (PCL_G) and PCL with the addition of gentamicin-modified montmorillonite (PCL_MMTG), which were subjected to microstructural, physicochemical (wettability) and mechanical tests. All membranes were also tested for water absorption, antibacterial activity and durability in environmental conditions. In addition, the studies of the release kinetics of gentamicin sulfate from PCL_MMG and PCL_G membranes were carried out in order to assess the possibility of extending the antimicrobial effectiveness.

## 2. Materials and Methods

### 2.1. Characteristics of the Starting Materials

Magnesium-aluminum montmorillonite (MMT) from Vanderbilt Company, Inc. under the trade name Veegum^®^F (pharmaceutical grade, purified and white) was used as a nanoadditive to obtain composite fibers. Montmorillonite has undergone an intercalation process using an active ingredient in the form of gentamicin sulfate (G) from Pharma-Cosmetic, Fagron, Poland (pharmaceutical grade, sterile).

Poly-ε-caprolactone (PCL, Sigma-Aldrich, Munich, Germany) with an average molecular weight (M_n_) of 80 kDa and a melting point of 58 °C was used to obtain composite and reference materials (from neat polymer). Dichloromethane (DCM, analytical grade Avantor SA, Gliwice, Poland) and N,N-dimethylformamide (DMF, analytical grade Avantor SA, Gliwice, Poland) were used as received as solvents.

### 2.2. Preparation of Samples

The preparation of montmorillonite intercalated with gentamicin sulfate (designated as MMTG) began with calcination of the starting montmorillonite at 200 ºC for 15 min. Then, it was intercalated with gentamicin sulphate in a round bottom flask in a water suspension at 50 ºC for 24 h with the use of a magnetic stirrer (MS 11 H, WIGO, Poznan, Poland). The obtained thick suspension was dried in a laboratory dryer (ZALMED SML 32/250, Warszawa, Poland) at 60 °C for 120 h, until a dry sediment was obtained, which was then pulverized in an agate mortar. The entire procedure has been described in detail in the authors’ previous works [28].

Electrospinning solutions were prepared using a mixture of DCM: DMF organic solvents mixed in a 4:3 *v*/*v* ratio. The fillers were prewetted with a mixture of solvents then used to dissolve the polymer. The wetting of MMT, MMTG or G in the DCM: DMF solution was carried out for about 24 h, and then PCL was introduced so that its content was 15% by weight. The amount of filler, based on the weight of dry polymer, was 2 wt.% In all composites. The mixture was mechanically homogenized for minimum of 24 h and additionally homogenized with ultrasound (1–3 min/60 Hz) in a water bath shortly before the electrospinning process (Sonix VCX 130PB apparatus) at room temperature. The production time of fibrous membranes, regardless of their composition, was 20 min, and the parameters of this process are presented in Table 1.

### 2.3. Methods

The initial powders of montmorillonite (MMT), calcined montmorillonite intercalated with gentamicin sulfate (MMTG) and pure gentamicin sulfate (G) were tested using the XRD technique. Measurements were made using the PANalytical Empyrean X-ray diffractometer with a copper lamp in the range of 3–70° of the 2θ angle, and the range of 5–12° of the 2θ angle was used for the analysis (PANalytical Empyrean X-ray, Cambridge, UK).

The particle size distribution of the MMT, MMTG and G powders was measured by the dynamic light scattering method using (DLS method) a Zetasizer Nano Series (Nano-ZS, Malvern Instruments, Worcestershire, UK). The measurements were carried out at 20 °C in distilled water in the case of MMT and MMTG powders and in ethyl alcohol in the case of gentamicin sulphate.

The observations of the morphology of the MMT and G starting powders and modified MMTG, as well as all fibrous membranes, were carried out using the NOVA NANO SEM 200 scanning electron microscope (FEI, Hillsboro, OR, USA). Before the observations, the powders were vaporized with a layer of carbon. The fibrous samples were coated with 10 nm gold layer using the rotary pump sputter coater (Leica EM ACE600, Wetzaler, Germany).

The physicochemical tests of the fibrous membranes were performed using a DSA 25 KRUSS goniometer, using high-purity water (UHQ) as the measuring liquid. The test was carried out under standard conditions, and the result is the average of 10 measurements with standard deviation for each of the materials.

In order to determine the water absorption capacity of fibrous nanobiocomposite membranes, their water absorption was tested. For the cut and weighed squares of nonwovens with dimensions of 4 × 4 cm, extracts were prepared in a 1:100 ratio of material to UHQ water, which were then incubated for 24 h at 30 °C. After this time, wet membrane mass (*m_k_*, g) was reweighed and then dried to obtained results of the dry sample mass (*m_p_*, g). Then, dates were used to determine the water absorption (*N*) according to the Equation (1):(1)$$N=\frac{m_{k}-m_{p}}{m_{p}}*100\;\%$$

Mechanical tests (tensile strength, Young’s tensile modulus and elongation at maximum stress) of the fibrous membranes were performed on a Zwick 1435 universal testing machine (Zwick Roell, Ulm, Germany) using a measurement speed of 5 mm/min. The tests were carried out on 5 × 60 mm strips cut from fibrous materials, and the working area was 40 mm.

Differential scanning calorimetry (DSC) was used to determine the melting point, glass-transition temperature and the degree of crystallinity of the tested materials using a Mettler Toledo STAR°SW 12.00 apparatus (Mettler-Toledo International Inc., Greifensee, Switzerland). The tests were carried out in the temperature range from −90 to 100 °C, and the test pieces weighed about 3.2–3.6 mg. The tests were performed under an inert atmosphere of nitrogen, using a standard temperature progression rate of 5 °C/min. The degree of crystallinity was determined on the basis of the Equation (2):(2)$$\chi =\frac{\Delta H_{f}}{w_{PCL\;}\times \Delta H_{100}}\times 100$$
where Δ*H_f_* is the experimental heat of fusion, *w_PCL_* is the PCL weight fraction and Δ*H*_100_ is the heat of fusion of 100% crystalline PCL (136.1 J/g) [46].

Studies on the kinetics of gentamicin sulfate release were carried out for the intercalated MMTG material as well as for the PCL_MMTG modified fibrous membrane, in accordance with the recommendations of the American Pharmacopoeia: Content of gentamicins sulfate [31]. A weighted portion of material (20 mg) was incubated in phosphate-buffered saline (PBS) (pH = 7.4) at 37 °C with the buffer volume to sample weight ratio being 1:5. Release rate studies were carried out over a period of 216 h; during which time, 1 mL of supernatant was withdrawn from the suspension, and the gentamicin sulfate content was determined by spectrophotometry using 330 nm wavelength (UV-VIS Cecil CE 2502, Corston, UK). The exact methodology of the determination was given in our earlier work [28,47], and the calculations were made on the basis of the Equation (3):(3)$$Cumulative\;release=\frac{V_{sample\;}}{V_{bath}}\times \left(P_{t-1}+P_{t}\right)$$
where *V_sample_* is the volume of sample withdrawn (ml), *V_bath_* is the bath volume (ml), *P_t_* is the percentage release at time *t* and *P_t−1_* is the percentage release previous to ‘*t*’.

Antibacterial tests were performed on fibrous membranes that were cut into squares with a diameter of 5 mm. The prepared materials were contacted with commercial strains of *Escherichia coli* bacteria (standard strain of Gram positive bacteria ATCC 25922) and *Staphylococcus aureus* (reference strain of Gram negative bacteria ATCC 29213). The Kirby–Bauer disk diffusion test was used, and the bacteria were grown in Petri dishes on MH agar medium (Mueller–Hinton agar, Biomaxima, Lublin, Poland) at 36 ± 1 °C for 24 h. The test procedure was performed according to the protocol of the American Society for Microbiology [48].

## 3. Results

### 3.1. X-ray Analysis of the Starting Montmorillonite Powders

XRD analysis was performed on montmorillonite powders before and after intercalation with gentamicin, and their results are shown in Figure 1.

In the measurement range shown, for pure montmorillonite (MMT), one reflex is observed at 7.2° of the 2θ angle, which corresponds to an interplanar distance *d_001(MMT)_* equal to 12.3 Å. The calcination of MMT and its subsequent intercalation with gentamicin sulfate (G) leads to a change in the position of the characteristic reflex in the MMTG material and thus a shift towards smaller angles to the position of 6.3°, which corresponds to the value of the interplanar distance *d_001(MMTG)_* equal to 13.9 Å (Table 2).

Shifting the value of *d_001_* towards the lower values of the 2θ angle means that there was an increase in the interplanar distance in MMT during the intercalation [47]. In the case of the diffractogram of the gentamicin sulphate powder (G), no reflections were noticed, which indicates the amorphous structure of this substance.

### 3.2. Grain Size Distribution of Montmorillonite Powders

Measurement of the particle size distribution (Figure 2) allowed for the assessment of the volume fractions of the particle size for the MMT, G and MMTG starting powders, as well as for the determination of their statistical distribution.

On the basis of the determined curves, it is possible to observe a clearly bimodal particle size distribution of the initial MMT, especially in the diameter ranges of 0.09–0.9 and 0.9–6.0 µm. There is also visible unimodal distribution of gentamicin sulfate particles in the range of 10–105 µm, as well as pseudobimodal distribution of MMTG particles, resulting from MMT intercalation with the active substance, where particles with diameters in the range of about 4–40 µm have the largest volume share. It was noted that this range is an intermediate range between the dominant values for MMT (lower) and for G (greater), which suggests that MMT intercalation occurred.

### 3.3. Morphology of the Starting Montmorillonite Powders

SEM micrographs along with EDS analysis taken for the starting MMT, MMTG powders and for the gentamicin sulfate powder are shown in Figure 3. The starting MMT is characterized by a lamellar grain shape with slightly jagged edges. The MMTG micrograph shows a greater development of the surface, and the EDS analysis shows a sulfur-derived peak, which is characteristic of gentamicin sulphate [42]. Gentamicin, on the other hand, has the form of spheres of various diameters, and the EDS analysis performed provides information about the presence of sulfur and oxygen, which proves the sulphated form of the drug.

### 3.4. Characteristics of the Microstructure of Nanobiocomposite Materials

The analyzed membrane materials were spun at a comparable time; hence, the thickness of the materials is on a similar level, in the range of 180–210 µm (Table 3).

The porosity of the membranes, determined by the gravimetric method at about 90%, is also comparable, and the discrepancy in the thickness of the membranes is directly related to the size of the fibers themselves (Figure 4). This correlates with the absorbability of membranes, which in the case of the filler with the addition of gentamicin sulphate (PCL_G and PCL_MMTG) is slightly higher than for the membranes: polymer (PCL) and modified with nonintercalated MMT (PCL_MMT).

The addition of the modifier increases the average fiber diameter from 380 µm for pure PCL to 396 µm for PCL_G and to 542 µm for PCL_MMTG. All fiber membranes are characterized by a unimodal fiber size distribution, with the most homogeneous population being pure polymer without nanoadditives (Figure 4). The presence of modifiers was confirmed by performing the EDS analysis, which includes the analytical elements for aluminosilicate (Al, Si, Ca, Na and Mg) but also for gentamicin sulphate (S).

### 3.5. Assessment of the Wettability of Composite Materials

Wettability studies have shown that the addition of a powder modifier to the fibers, either in the form of gentamicin sulfate (G) or in the form of modified (MMTG) or unmodified (MMT) aluminosilicate, causes a decrease in the hydrophobicity of the polymer membrane (Figure 5).

While the value of the contact angle for a neat PCL membrane is about 120°, the addition of MMT reduces this value by about 15° (to 105°). On the other hand, gentamicin sulphate added directly to the spinning solution has the strongest effect on the increase in wettability of the fibrous membrane, and the contact angle measured drops by 22° and 24° for the PCL_G and the PCL_MMTG membrane, respectively. Therefore, it seems that both the decrease in wettability and the increase in water absorption (Table 3) of the membranes is related to the addition of gentamicin sulphate.

### 3.6. Mechanical Properties of Nanobiocomposite Materials

The mechanical properties of nanobiocomposite membranes are summarized in Table 4, and the analysis of the results shows that the modulus of elasticity and tensile strength strongly depend on the type of modifier added to the polymer solution at the stage of producing fibrous membranes.

It is clearly visible that with the addition of MMT and MMTG, the average fiber thickness and the tensile strength of the PCL_MMT and PCL_MMTG membranes increase. The fibers in the membranes are heterogeneous and arranged in different directions; hence, the fluctuations in the value of force deformation shown in Figure 6. The addition of aluminosilicate also increases the stiffness of the nanobiocomposite material, and the highest value of Young’s modulus was shown by PCL_MMT membranes, for which the highest elongation to break was reported during tensile test. The obtained results are influenced by the morphology of the fibers, possible defects, e.g., pores, exposed additives or additives that have not been sufficiently wetted by the polymer during the electrospinning process, and which can be considered as a type of inclusions. Such an example seems to be the PCL_G membrane with the lowest value of Young’s modulus and elongation in relation to the nanobiocomposite materials PCL_MMT and PCL_MMTG.

### 3.7. Thermal Studies of Nanobiocomposite Materials

The thermal properties of nanobiocomposite electrospun membranes were compared with the reference membrane made of neat PCL, and the obtained results are summarized in Table 5. No changes in the melting temperature of the fibers were observed after the addition of clay (MMT and MMTG) or gentamicin sulphate itself. On the other hand, the glass-transition temperature changed (decrease by about 10 °C) and thus the theoretically determined crystallinity of PCL_MMT PCL_MMTG and PCL_G fibers. The observed increase in the crystallinity to about 55% (which is 10% more than the determined crystallinity of pure fibrous PCL) proves that during the electrospinning conditions that were used, structural changes might occur, causing an increase in the order of the polymer chains, as long as there is a suitable modifier (e.g., MMT, MMTG or even G alone).

### 3.8. Assessment of Antibacterial Properties and Kinetics of Gentamicin Sulfate Release from Nanobiocomposite Materials

The assessment of antibacterial properties was based on the observation of the zones of inhibition of bacterial growth (*S. aureus* and *E. coli*) after contact with fibrous materials. The performed microbiological tests were aimed at unequivocally confirming the presence of the active substance, i.e., gentamicin sulphate and determining the kinetics of the active substance release (Figure 7). The largest zone of inhibition of bacterial growth was observed for the PCL_G and PCL_MMTG membranes against Gram-positive *S. aureus* bacteria (the zones of inhibition were 25 and 23 mm, respectively). These materials showed lower efficacy against Gram-negative *E. coli* bacteria, and the corresponding zones of growth inhibition were in this case, 24 mm for PCL_G and 23 mm for PCL_MMTG (Table 6). For the pure PCL membrane, there was a minimum zone of inhibition of growth against *E. coli* (14 mm) and larger zone of inhibition of growth against *S. aureus* (15 mm).

The cumulative curves shown in Figure 8 show that the most effective release of gentamicin sulfate occurs when it is intercalated into the interlayer spaces of aluminosilicate (MMTG powder). In this case, the antibiotic remains unbound in the polymer matrix, and after its initial burst, the release is visibly stabilized. The release of gentamicin sulfate from the polymer matrix (PCL_G) is significantly slower than that of MMTG, and there is no typical range for a burst of active compound, and the accumulation curve is practically linear. A similar course, but with lower levels of released active ingredient, takes place in the PCL_MMTG membrane. Here, the effect of the presence of sulphate is also visible after the 3rd day, after which the sulphate concentration increases significantly, and the shape of the curve from the 4th day is linear, meaning a successive release of the active compound. The proposed release time of 216 h is too short to achieve the complete dissolution of montmorillonite from the interlayer spaces.

## 4. Discussion

Structural analysis of MMT and MMTG powders confirms that calcination may be a step facilitating intercalation of layered aluminosilicate with gentamicin sulfate. Evidence of the intercalation occurrence is the change in the position of the main peak *d_001_* for MMT and its shifting towards the lower values of the 2θ angle (Table 2). As a result, the increasing MMTG interplanar distance confirms the effectiveness of the proposed modification method. The earlier works of the authors [28,47] on the possibility of modifying the nanoclay indicate that the two-stage chemical and thermal treatment processes facilitate the introduction of active agents into the interlayer spaces of the aluminosilicate. As a result, not only the structural parameters change but also the particle size distribution of the clay, which, like the distribution of MMT, is still pseudodimodal; however, the particle size is an intermediate value between MMT and G (Figure 2). Chemical and thermal treatment also change the morphology of MMT platelets. After MMTG modification, they become finer and disintegrated, which may indicate not only the intercalation process, but also a partial exfoliation of MMT packets (Figure 3). In the literature, there are known methods of intercalation and exfoliation of MMT clays, which are usually based on the introduction of an aggressive chemical agent (np. N-dimethylacetamide, hydrogenated tallow alkyl), which, under certain process conditions, leads to the expansion of the interlayer space and sometimes to the complete disintegration of the packets and total exfoliation of MMT [33,49,50].

Assuming the use of intercalated clay for the electrospinning process, it should be borne in mind that the intercalation process itself facilitates the homogenization of the filler in the polymer matrix [51]. When choosing a nanofiller, many authors are suggested by the chemical treatment, wettability or the behavior of particles in the system of solvents used for electrospinning [51,52,53,54]. In our case, the indication for further experiments was the sustained homogeneous distribution of MMTG in the DCM:DMF mixture after 24 h of mechanical wetting and sonication. Our proposed method of processing the MMT filler and its compatibility with the PCL matrix confirms the possibility of obtaining PCL/MMTG fibrous membranes. 

As it is known, taking into account three groups of parameters, which are to guarantee the effective course of electrospinning, is a compromise between the conditions in the solution, the conditions of the process and the environment in which the process takes place. The electrospinning process takes advantage of the potential difference between the earthed collector and the tip of the nozzle, which creates a strong electric field. In our case, an additional disturbance of the solution was the presence of insulating ceramic nanoparticles, which made it necessary to increase the voltage from 15 to 18 kV and the speed of feeding the polymer with the modifier to the nozzle of the device. Leaving the polymer to low flow caused the clogging of the nozzle or the appearance of defects in the form of polymer droplets (beads) on the manifold. On the other hand, a too high feeding speed at the assumed needle-collector distance (typically 10 cm for PCL) caused the sticking of the polymer fibers; as a consequence of which, a polymer film was formed instead of the morphology of the fibers. Ultimately, the remoteness of the nozzle and the use of temperature and humidity control with the proposed solvent system (volatile DCM and less volatile DMF) allowed for the production of good-quality fibers with smooth surface, free of defects (pores) and free of free MMT or MMTG nanoparticles. The developed conditions for the electrospinning process allowed for the production of membranes with comparable thickness, porosity and water absorption. The increase in fiber thickness in the PCL_G, PCL_MMT and PCL_MMTG membranes proves the effective modification of the fiber, which is confirmed by the EDS analysis of nanocomposite fibers (Figure 4), in which, in addition to the analytical elements for the polymer (oxygen, carbon), there are silicon, aluminum in PCL_MMT and silicon, aluminum and sulfur in PCL_MMTG. Importantly, there were no changes in the morphology of single fibers in the form of, for example, defects, so often shown in the literature when the fillers are silica, titania, talc, starch, carbon nanotubes and graphene oxides [54,55,56]. The fiber size distribution depends on the size of the modifier—the smallest when it is gentamicin sulphate, the largest when it is clay (MMT) and modified clay (MMTG). Due to the presence of the filler in the fiber, it is possible to reduce the roughness effect; unfortunately, the specific microstructure of fibrous membranes usually increases the hydrophobicity of such materials, placing them on the border of highly hydrophobic and sometimes even superhydrophobic materials (the contact angle above 150°) [41]. The contact angle of the PCL_MMTG membrane also takes an intermediate value between PCL_G and PCL_MMT, which may indicate a homogeneous distribution of the filler in the fiber, which, in combination with the fiber size distribution, reduces the degree of membrane wettability (Figure 5).

The high value of the contact angle is due to the rough microstructure of the membrane due to the presence of fibers. The wetting of the material in this form is difficult due to the greater development of the surface and the longer time necessary for water to penetrate the irregular surface of the nonwoven fabric; hence, the contact angle for PCL is the highest.

There are reports in the literature on the influence of MMT on PCL matrices, for example, and, for example, Sołtysiak et al. [57] show that even PCL-cast films, into which MMT particles have been introduced, gain roughness, and the presence of MMT particles influences the ease of formation of polymer spherulites, and MMT itself can act as an initiator of nucleation of the polymer chain. In the case of our fibrous material, the effect of an increase in the fiber diameter due to the presence of MMT filler in its volume can be observed (Figure 4), and the increase in the fiber diameter and wider distribution of the fiber size is the effect of ‘smoothing’ the surface of the nonwoven fabric, and therefore better and faster penetration of the surface by drops of water (easier for it to flow), which results in a lower value of the contact angle. Further drops in the contact angle may be caused by the presence of gentamicin not only between the montmorillonite layers but also by the presence of gentamicin sulfate salt particles on the surface of the powder added to the electrospinning. This assumption is confirmed by the almost identical value of the contact angle for the PCL_G and PCL_MMTG materials.

The use of a slow speed manifold makes the fiber system random. This fiber microstructure determines specific mechanical properties, different from those observed in unidirectional fiber systems [58,59,60,61]. By analyzing the course of the curves on the basis of the tensile-strength–elongation relationship, it can be concluded, based on theoretical considerations, that PCL_MMT membranes are characterized by the highest strength, and thus the strength of interfiber contacts (the so-called complete fusion). They are probably due to the presence of solvent in the nanofillers (on the surface, MMT flakes rather into spaces of MMT) and slower evaporation of solvents from the fibers (hence, the fibers stick together). In theoretical works, it is assumed that such a fiber-to-fiber fusion is a chemical bond, while the Van der Waals forces are responsible for the fiber–fiber interactions of the nonmelted type [61]. If the nanofiller is introduced in the form of MMT intercalated with gentamicin sulphate, the interlayer distance increases, but with it, the degree of packing of this space also increases, making it difficult for DCM:DMF solvents to penetrate, which translates into a lower value of fiber fusion, and thus a lower value of resulting Young’s modulus (Table 4). The data published by Hao Yi et al. show that the MMT surface interacts weakly with water molecules, whereas exchangeable cations adsorbed on the MMT surface have a very strong affinity for water molecules. In fact, exchangeable cations are electrostatically adsorbed on the MMT surface in the form of a hydration shell with a cation core and an envelope of water molecules [62]. In our case, if there are sulfate ions on the flake surface, then they are surrounded by water, and more importantly, this is the property that plays a dominant role in giving the hydrophilic character to the MMT surface. The higher number of exchangeable cations and stronger hydration capacity of cations on the surface of MMT play a dominant role in making it hydrophilic.

The observed behavior additionally induces an intramolecular change in the polymer chain, which becomes more crystalline in the presence of the filler (Table 5). It is true that this behavior is quite often observed among nanocomposite materials [51,63], but less often described when electrospinning from a solution is used as a manufacturing method. In research on electrospun fibers, no significant changes in the degree of crystallinity were reported, which is explained by the too short duration of the process and difficulties in reorienting the polymer chain on the path between the nozzle and the collector [53]. In the case of PCL_MMT and PCL_MMTG nanocomposite fibers, it proves a high dispersion of fillers during the electrospinning process, which may facilitate the formation of crystalline regions during the preparation and result in an increase in the degree of crystallinity by about 10%.

The final confirmation of the presence of gentamicin sulphate in the fibrous membrane and the possibility of regulating its release time are the results of microbiological tests and the determination of the rate of release of this active substance from the polymer membrane (Table 6, Figure 8). The high activity of the PCL_MMTG membrane, comparable to that of the PCL_G membrane, both in contact with *S. aureus* and *E. coli*, indicates that both of these materials have an antibacterial effect. Gentamicin sulfate has a bactericidal effect both in the resting and growth phases of aerobic Gram-negative bacteria. It acts on the cytoplasmic membrane and ribosomes, leading to changes in the translation of bacterial proteins [46].

There are many reports in the literature on materials based only on intercalated nanoclays. In some cases, this solution works because it allows for a controlled release of catalysts initiating chemical reactions, salts for complexing contaminants or drugs from such a carrier [64,65,66]. The release process of such substances can be slowed down or more controlled by introducing intercalated nanoclays into the polymer matrix. Here, however, their appropriate form is required, enabling the activation of active substances, although, unfortunately, in many cases the form of the nanoparticle itself is sometimes an unacceptable form due to its size and not fully foreseen behavior when such a particle enters the microcomposition. In such cases, porous microstructures, including fibrous membranes, work well—even more so as studies show that the form of electrospun fibers is suitable for the controlled release of drugs and other active substances, and the form of the membrane facilitates the application of such a material [67,68,69].

With a view to extending the membrane performance, it seems that the gentamicin sulfate release is more advantageous in the PCL_MMTG membrane—about 6 % of the antibiotic is released within 1 h, while 24% is released from MMTG powder at the same time. Most likely, the kinetics of gentamicin release from fibrous membranes is influenced by the wettability of the membrane itself, and the greater the wettability, the faster the membrane material penetrates through the medium and its release faster—as is the case in PCL_G. The layered aluminosilicate modified with gentamicin sulfate in an aqueous solution, which is a phosphate buffer, is well penetrated by water and the ionic components of the buffer, thereby removing sulfate from the MMT gallery, resulting in an increase in the concentration of sulfate in the solution (Figure 8). In turn, the introduction of the intercalated filler into the polymer matrix protects it against strong water penetration, and the polymer layer protects the active compound in the aluminosilicate gallery. As a consequence, there is a slower release of gentamicin sulfate for the PCL_MMTG material, which is visible in the form of a lower concentration observed after 6 and 216 h of observation. Gentamicin sulfate is released faster in the system in which it is directly covered by the polymer layer and is not bound by electrostatic interactions with the carrier, which is the modified MMTG (release intermediate). The strongly developed surface of MMT modified with gentamicin sulfate (MMTG) releases the antibiotic more slowly, as described in previous studies. They proved that gentamicin sulfate is bound both superficially and in volume (intercalates into the MMT gallery space). In such a system, there is a slower release of the antibiotic from the PCL fibers (because there is less of it on the flap surface) compared to the unbound pure salt present in the PCL_G fibers.

The lower the wettability of the membrane (PCL_MMTG), the slower the release of gentamicin sulfate into the medium takes place, and this time is further lengthened by the antibiotic confinement in the interlayer spaces of MMT. Thus, it can be concluded that the formation of connections of the intercalated active substance MMT with the polymer matrix leads to an extended release time of the active substance from this type of composite materials, thus making it possible to sustain an antibacterial function over a more effective period of time.

## 5. Conclusions

The conducted research shows the effectiveness of the electrospinning method to obtain both PCL-based nanobiocomposite fibers modified with MMT-based aluminosilicate and with intercalated gentamicin sulphate-MMTG aluminosilicate. The effectiveness of intercalation was confirmed by the conducted structural study and application tests of gentamicin sulphate release as well as by microbiological tests. The results of microbiological tests confirmed the antibacterial activity of all the materials obtained. The electrospinning technique can be also effectively used to obtain PCL_MMT and PCL_MMTG nanobiocomposite fibers with improved breaking strength and increased Young’s modulus compared to materials made only of polymer fibers, provided that a high filler dispersion in the spinning solution is obtained. The presented PCL_MMT, PCL_MMTG or MMT_G nanobiocomposite membranes can find potential application both in the food industry (packaging) and in biomedicine, in the form of single- or multi-layer systems.

## Figures and Tables

**Figure 1 materials-14-06905-f001:**
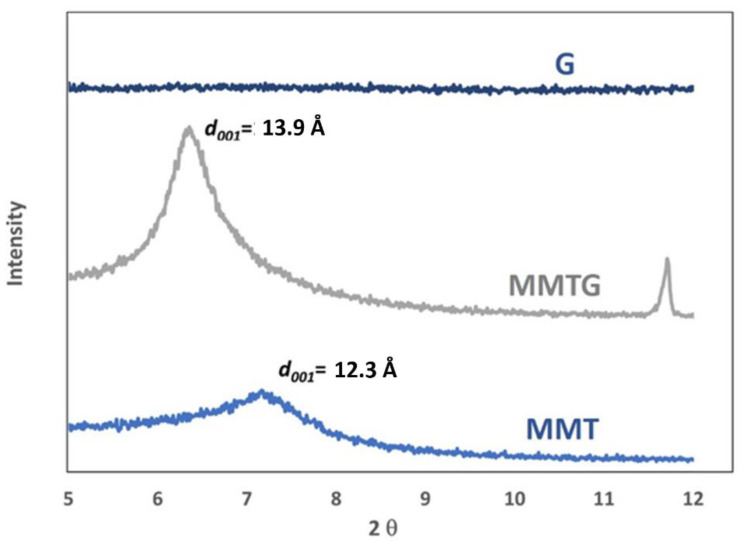
X-ray diffractograms of the tested powders (G, MMT, MMTG).

**Figure 2 materials-14-06905-f002:**
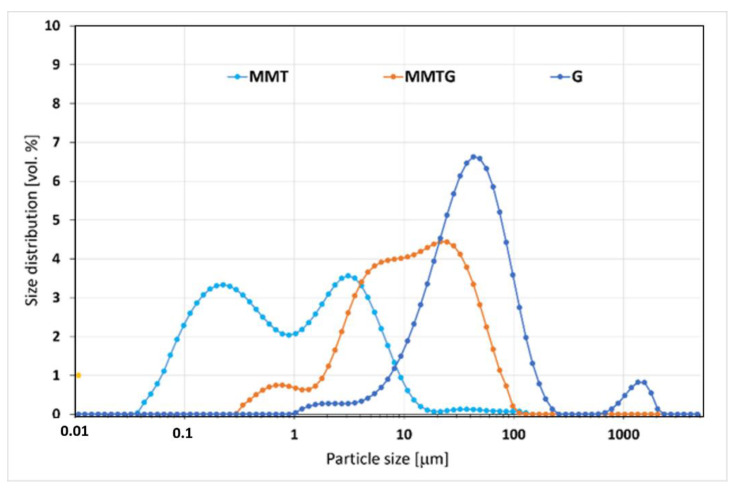
Grain size distribution of MMT, G and MMTG powders.

**Figure 3 materials-14-06905-f003:**
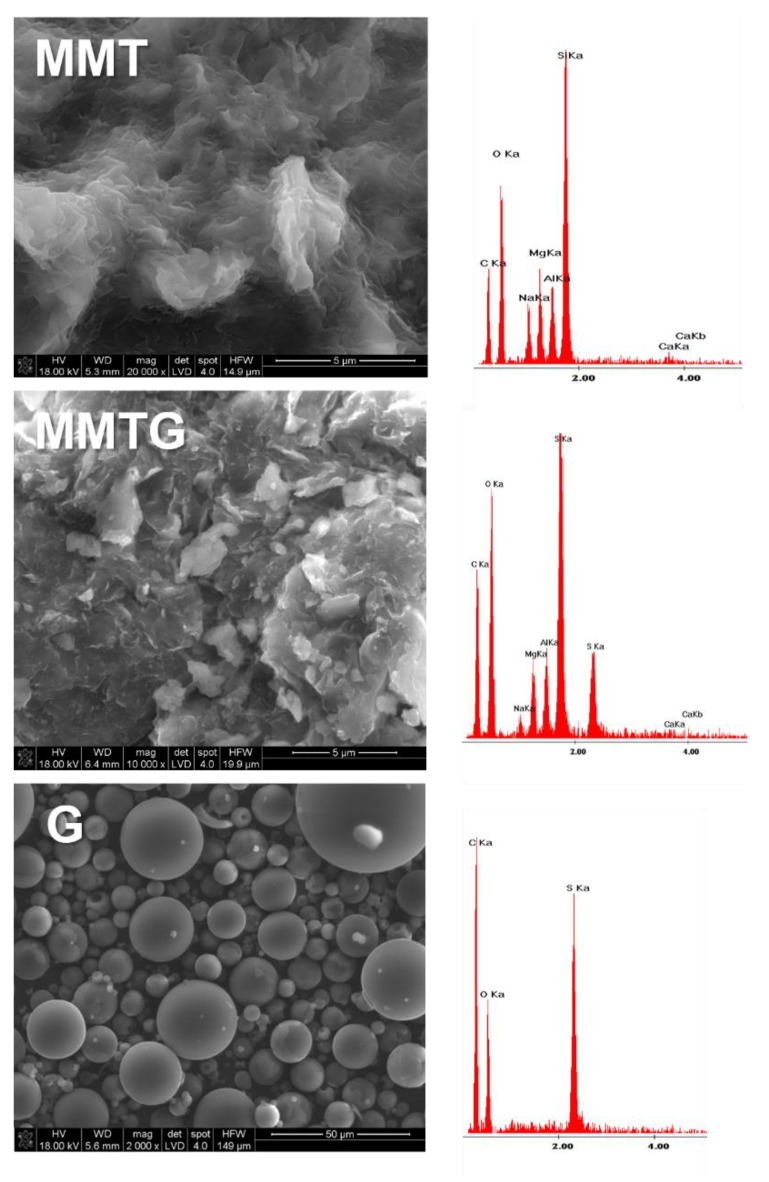
SEM image of the MMT, MMTG and G powders with EDS analysis (average analysis of the observed microarea).

**Figure 4 materials-14-06905-f004:**
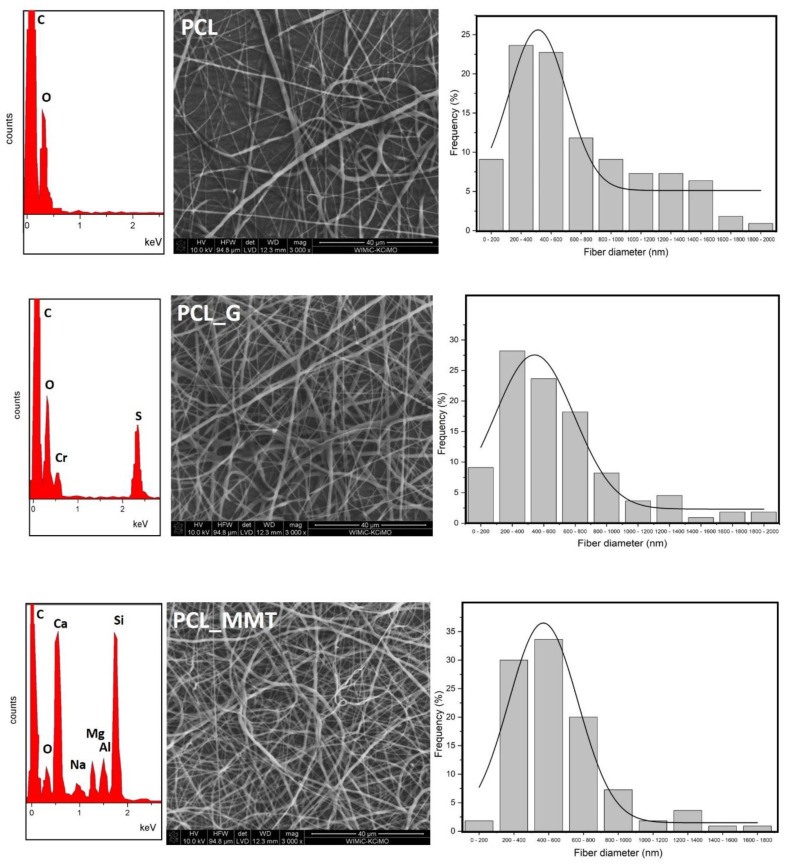
Microstructure of PCL_G, PCL_MMT and PCL_MMTG nanobiocomposite fibers and PCL polymer fibers with the fiber size distribution determined on the basis of SEM with EDS (average analysis of the observed microarea).

**Figure 5 materials-14-06905-f005:**
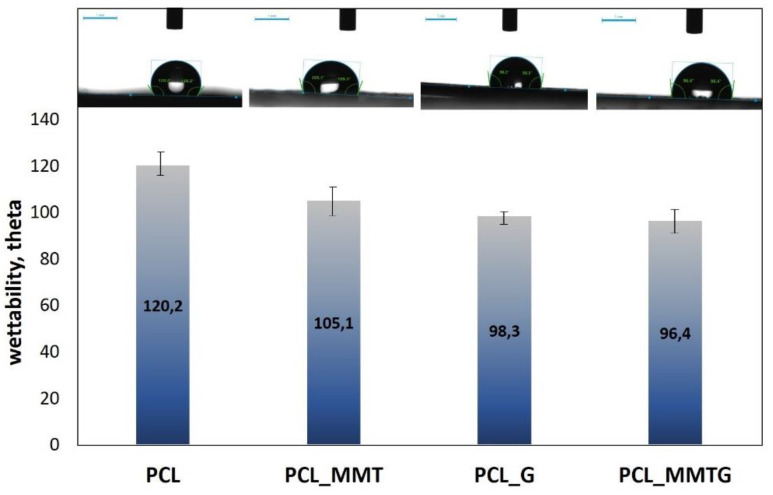
Wettability of nanocomposite fibrous membranes: PCL_MMT, PCL_MMTG and PCL_G vs. wettability of PCL polymer membrane.

**Figure 6 materials-14-06905-f006:**
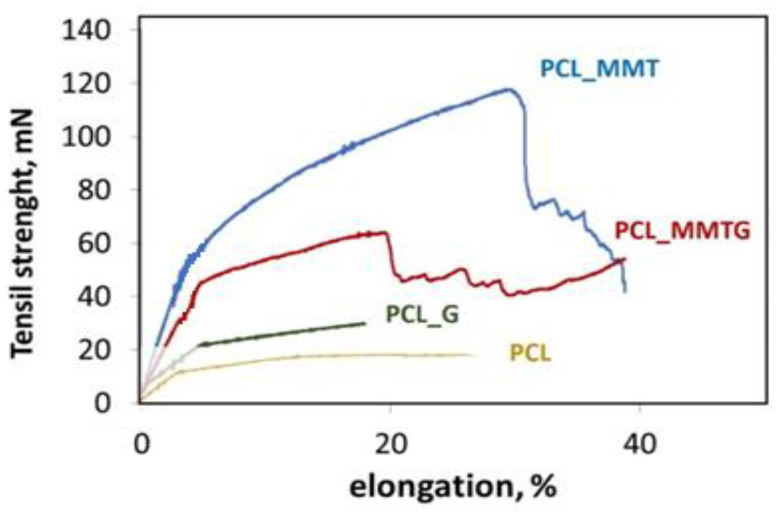
Force-elongation curves recorded during mechanical tests of nanobiocomposite membranes: PCL_MMT, PCL_MMTG and PCL_G and the reference PCL polymer membrane.

**Figure 7 materials-14-06905-f007:**
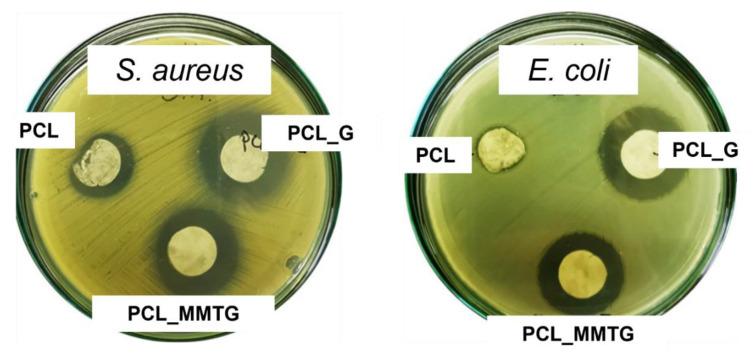
Antimicrobial tests results of PCL, PCL_G and PCL_MMTG materials against *S. aureus* and *E. coli*.

**Figure 8 materials-14-06905-f008:**
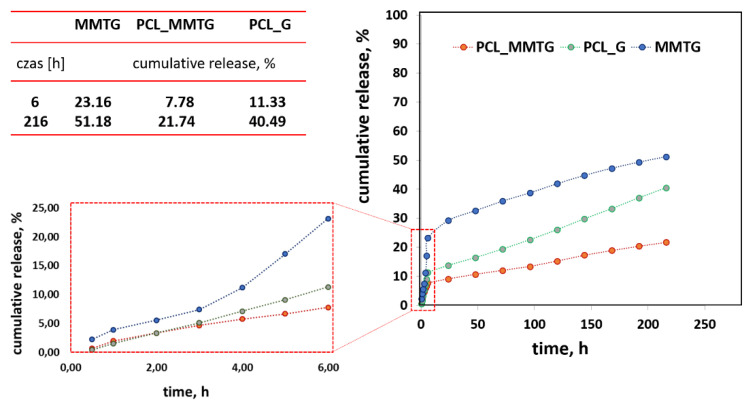
The rate of antibiotic release from the MMTG powder and PCL_MMTG and PCL_G materials during a 216 h in vitro study in PBS of pH = 7.4 at 37 °C.

**Table 1 materials-14-06905-t001:** Technical data and conditions for obtaining materials by electrospinning.

Materials	PCL	PCL_MMTG	PCL_G	PCL_MMT
Ultrasound homogenization, min/Hz	-	3/60	1/60	1/60
Chamber humidity, %	20	25	21	21
Chamber temperature, °C	21	20	21	20
Voltage, kV	15	18	18	16
Needle-collector distance, cm	10	11	12	12
Collector rotational speed, cm/min	5			
The flow rate of the solution stream, mL/min	1.2	1.8	1.0	1.5

**Table 2 materials-14-06905-t002:** The values of the basal interlayer spacing for MMT and MMTG powders calculated based on the position of the main peak.

Powder	Main Peak Location, 2θ	Interlayer Distance *d_001_*, Å
MMT	7.2	12.3
MMTG	6.3	13.9

**Table 3 materials-14-06905-t003:** Characteristics of fibrous membrane materials.

Material	Membrane Thickness, µm	Average Fiber Size, µm	Total Porosity of the Membrane, %	Water Absorption of the Fibrous Membrane, %
PCL	206 ± 12	461 ± 75	92	59
PCL_G	178 ± 15	561 ± 67	89	66
PCL_MMT	210 ± 27	684 ± 135	86	62
PCL_MMTG	196 ± 24	591 ± 63	90	70

**Table 4 materials-14-06905-t004:** Mechanical properties of nanobiocomposite materials.

Material	Tensile Strength MPa	Young’s Modulus MPa	Elongation at Break %
PCL	0.029	0.54	23
PCL_G	0.037	0.56	19
PCL_MMT	0.112	1.22	31
PCL_MMTG	0.078	0.84	29

**Table 5 materials-14-06905-t005:** Summary of DSC results for the tested PCL fibrous material and PCL-based nanobiocomposites.

	T_g_, °C	T_m_, °C	Δ_f_, J/mol	χ, %
PCL	−53.7	62.8	−71.2	45.3
PCL_MMT	−63.7	64.7	−87.1	55.6
PCL_G	−61.2	63.9	−88.2	56.2
PCL_MMTG	−65.2	63.8	−87.5	55.7

**Table 6 materials-14-06905-t006:** Growth inhibition zones [mm] around the tested nonwovens after contact with Gram-positive (*S. aureus*) and Gram-negative (*E. coli*) bacteria.

Material	The Diameter of the Zone of Inhibition of Bacterial Growth, mm	
	*S. aureus*	*E. coli*
PCL	15	14
PCL_G	25	24
PLC_MMTG	23	23

## Data Availability

Not applicable.

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
