# Peer review of "Effects of Montmorillonite and Gentamicin Addition on the Properties of Electrospun Polycaprolactone Fibers"

_materials, 2021, doi:10.3390/ma14226905_

Round 1
Reviewer 1 Report
Dear Editor:
Thank you for the opportunity to review the manuscript " Effect of montmorillonite addition on
the properties of electrospun polycaprolactone fibers" submitted to materials by Stodolak-Zych et
al. This manuscript (materials-1361097) is prepared well, and I believe that the readers can have
some benefits from this study. It is novel and a topic of interest to researchers in related areas.
There’s still some minor revision that I suggest for its publication as seen below.
1) Please remove those statements from the manuscript: “Error! Reference source not found”.
2) Explain more about the potential applications of the fabricated nanocomposites.
3) Add some numerical results in the abstract section.
4) Discuss in detail the water contact angle measurement procedure in the manuscript.
5) Discuss in detail the nanocomposite electrospinning process i. e. feed rate, voltage, etc.
6) There is some clumsy English grammar/phrasing throughout the manuscript. English/
grammar of the manuscript can be improved before the publication.
7) Overall, the manuscript seems good and can be considered for publication in your
journal.
Author Response
Dear Reviewer,
Thank you for careful reading our manuscript entitled: Effects of montmorillonite and gentamicin addition on the properties of electrospun polycaprolactone fibers (Materials-1361097). We hope that the answers below will provide a better explanation of our manuscript.
Q1. Please remove those statements from the manuscript: “Error! Reference source not found”.
Thank you for your comment, of course all sentences ‘Error! Reference source not found’ indicating an editing defect (automatic literature list) have been removed and corrected. Corrected and highlighted in yellow in the text.
Q2. Explain more about the potential applications of the fabricated nanocomposites.
The suggested nanocomposite materials based on one of the aliphatic polyesters which is PCL can be used in a wide range of packaging industry. In literature there are many solutions showing preparation of such material. Possibility of application of nanocomposites with MMT in this scope results from their better mechanical and thermal properties [1-5]. Additionally, based on the processing capabilities of the die, these plastics can be formed using extrusion, moulding or casting [2-3]. Unfortunately, in all approaches there is a problem with homogeneous distribution of nanofiller. If methods based on the thermoplastic properties of the polymer (e.g. PLA) are used to form the nanocomposite, a precalcined clay is usually added as a modifier. Such filler preparation facilitates dispersion during the homogenization process. A frequent additive facilitating clay compatibilization are monomers e.g. PCL monomer [6]. On the base of obtained results of mechanical properties investigations it was stated that addition of clay and PCL composite significantly increases the strength and stiffness of PLA/PCL nanocomposites with MMT. Additionally, storage modulus of the material increases, which can be used to predict the behavior of the material during operation (stretching) and under temperature influence. The casting method is based on dissolution of polymer in an organic solvent and improvement of wettability of clay nanoparticles during the pre-dispersion process [7]. In such systems, clay is modified with ammonium salts often based on commercially available products such as Cloisite Na, Cloisite 30B and Cloisite 20A. In order to improve the filler-polymer matrix interaction, which results in structural (increased degree of crystallinity and thermal stability) and mechanical (strength, Young's modulus) changes [8]. A number of solutions in the field of polyester-based nanocomposite materials aim at giving an antibacterial function and/or inducing a controlled degradation process of the material. In this field, the additives are mainly oxides of copper, zinc or titanium or nanoparticles of zinc, titanium [6-9]. The suggested fibrous nanocomposite solution with gentamicin intercalated MMT can act as a multifunctional nanocomposite material; antibacterial and degradable under natural conditions.
Another application of nanocomposite systems based on aliphatic polyesters and nanoclay is biomedical engineering. Popularity of the solutions in this area, similarly as in the case of packaging, results from a number of processing methods made possible by polymeric matrices as well as the possibility of modification of the filler itself; with antibiotics and monomers [9-11]. Our previous work shows that nanocomposites of PLA and modified clay can be prepared by injection molding, in which the amount of filler determines the durability of the material in highly hydrated conditions (the more filler, the faster the material loses molecular weight and mechanical properties) [12]. On the other hand, gentamicin-modified PLA/MMT cast materials show strong antibacterial properties against gram-positive bacteria, i.e. E.coli [13]. Application of the salt leaching method, in turn, allows to obtain porous substrates whose pore size depends on the porogen (salt) concentration. The introduction of MMT and modified MMT leads to the gradient character of the substrate due to the sedimentation processes that take place during the drying process [14]. Based on such experimental experience, the prepared fibrous materials can also represent an attractive material from the point of view of tissue engineering as well as regenerative medicine; a porous substrate/membrane that can have an antibacterial character as well as a controlled degradation time.
[1] Natalia V. Pogodina et al., Processing and characterization of biodegradable polymer nanocomposites: Detection of dispersion state, Rheol Acta (2008) 47, pp.543–553.
[2] A.K., Mohanty, et al., M., Nano-reinforcement of biobased polymers—the hope and reality, Polym. Mater Sci. Eng., 2003, 88: pp.60–1.
[3] Pinnavaia TJ, et al., Polymer – clay nanocomposites, Wiley series in polymer science; 2000
[4] P. Das, J. M. Malho, K. Rahimi, F. H. Schacher, B. C. Wang, D. E. Demco, A. Walther, Nat. Commun. 2015, 6, 5967
[5] B. M. Holzapfel, J. C. Reichert, J. T. Schantz, U. Gbureck, L. Rackwitz, U. Noth, F. Jakob, M. Rudert, J. Groll, D. W. Hutmacher, Adv. Drug Delivery Rev. 2013, 65, 581
[6] Sinha, R.S. et al., Structure-property relationship in biodegradable poly(butylenesuccinate)/ layered silicate nanocomposites. Macromolecules36, 2003, pp.2355-2367
[7] Jong-Whan Rhim et al., Effect of PLA lamination on performance characteristics of agar/κ-carrageenan/clay bionanocomposite film, Food Research International 51 (2013) pp. 714–722.
[8] R.Umamaheswara Raoa et al. Enhancement of Mechanical Properties of PLA/PCL (80/20) Blend by Reinforcing with MMT Nanoclay, Materials Today: Proceedings 18 (2019) 85–97
[9] K. Fukushima, et al., Nanocomposites of PLA and PCL based on montmorillonite and sepiolite, Materials Science and Engineering C 29 (2009) pp.1433–1441
[10] Bujok s et al., Sustainable microwave synthesis of biodegradable active packaging films based on polycaprolactone and layered ZnO nanoparticles Polymer Degradation and Stability 190, 2021, 109625
[11] G. Gorrasi et al., Vapor barrier properties of polycaprolactone montmorillonite nanocomposites: effect of clay dispersion, Polymer 44 (2003) 2271–2279
[12] A. Rapacz-Kmita et al., The wettability, mechanical and antimicrobial properties of polylactide/montmorillonite nanocomposite films Acta of Bioengineering and Biomechanics 19 (4) 2017 doi; 10.5277//ABB-00820-2017-02
[13] A. Rapacz-Kmita, M.. Bućko, E. Stodolak-Zych, M. Mikołajczyk, P. Dudek, M. Trybus, Characterisation, in vitro release study, and antibacterial activity of montmorillonite-gentamicin complex material, Mater. Sci. Eng. C. Mater. Biol. Appl. 2017, 70 (Pt 1), 471–478. https://doi.org/10.1016/J.MSEC.2016.09.031
[14] MT Albdiry, et al., A critical review on the manufacturing processes in relation to the properties of nanoclay/polymer composites, Journal of Composite Materials 4, 2014, pp 1-24
Q3. Add some numerical results in the abstract section.
Abstract has been corrected and completed with the data requested by the reviewer. Corrected and highlighted in yellow in the text.
Q4. Discuss in detail the water contact angle measurement procedure in the manuscript.
An additional translation regarding wetting angle measurements has been completed in the manuscript.
The high wetting angle value is due to the rough microstructure of the membrane resulting from the presence of fibers. Wetting of the material in such a form is difficult due to the higher surface expansion and longer time necessary for water to penetrate the irregular material surface [1]. Therefore, the wetting angle for PCL is the highest. The phenomenon which occurs during wetting of submicron and nanometric surfaces is known in literature as Cassie's wetting model (Figure 1).
Figure 1. Wetting models and working mechanisms of surfaces [2]
The statement above is supported by the literature. Sołtysiak et al. show that even cast PCL films into which MMT particles were introduced gain in roughness - the presence of MMT particle influences the ease of polymer spherulites formation and MMT itself can act as a factor initiating nucleation of polymer chain. In the discussed case of fibrous material, we can observe the effect of increase in the fiber diameter due to the presence of MMT filler in its volume (Figure 4 in the manuscript). An increase in the fibre diameter and a wider fibre size distribution is an effect of 'smoothing' of the nonwoven surface, and thus a better and faster penetration of the surface by water droplets (easier dispersion), which results in a smaller value of the wetting angle. Further decrease in contact angle can be caused by the presence of gentamicin not only between montmorillonite layers but also by the presence of particles of genramycin sulfate salt on the surface of powder added to electrospinning. This assumption is confirmed by almost identical value of contact angle for PLA_G and PLA_MMTG materials.
[1] Anish Tuteja, Wonjae Choi et al. Design parameters for superhydrophobicity and superoleophobicity, MRS Bulletin 2008 33 (8), 752-758
[2] D. Zhang, L. Wang, H. Qian, et al. Superhydrophobic surfaces for corrosion protection: a review of recent progresses and future directions. J Coat Technol Res 13, 11–29 (2016)
[3] E. Sołtysiak, M. Błażewicz, influence of nanoparticles on physical and chemical surface material properties, Eng Biomat, 92, (2010), 30-3
Q5. Discuss in detail the nanocomposite electrospinning process i. e. feed rate, voltage, etc.
As is well known, consideration of the three groups of parameters to ensure successful spinning is a compromise between solution conditions, process conditions, and the environment in which it occurs. The electrospinning process is the exploitation of the potential difference between the grounded collector and the nozzle end, which results in the generation of a strong electric field. In our case, an additional disturbance to the solution that the electric field had to deal with is the presence of ceramic nanoparticles (insulators). It was therefore necessary to increase the voltage from 15 to 18kV and the speed of feeding the polymer with modifier to the nozzle of the device. Leaving a low polymer flow rate caused the nozzle to pivot or the appearance of defects in the form of polymer droplets (beads Figure 1a) or thick polymer plumes deposited on the collector (Fgura 1b). On the other hand, too high delivery speed at the assumed needle-collector distance (typical for PCL 10cm) causes polymer fibers to stick together, resulting in polymer film instead of fiber morphology (Fig. 1d). Moving the nozzle away and using temperature and humidity control with the proposed solvent system (volatile DCM and less volatile DMF) enabled the formation of a nonwoven film (Figure 1c). The developed conditions were optimized until a satisfactory morphology of fibers included in the manuscript was obtained. Data of the process is given in Table 1 in the manuscript.
Figure 2. Defects on trawls appearing during optimization process of electrowinning of trawls based on PCL_MMT and PCL_MMTG: beads (a), polymer spitting (b), fiber merging (c,d).
Q6. There is some clumsy English grammar/phrasing throughout the manuscript. English/
grammar of the manuscript can be improved before the publication.
The manuscript has been submitted for grammatical proofreading by a native speaker with a technical background. We hope that the current language will facilitate the reception of the manuscript. Corrected and highlighted in yellow in the text.
Reviewer 2 Report
Please see attached Word Document

Author Response
Dear Reviewer,
Thank you for careful reading our manuscript entitled: Effects of montmorillonite and gentamicin addition on the properties of electrospun polycaprolactone fibers (Materials-1361097). We hope that the answers below will provide a better explanation of our manuscript.
Due to more than one hundred comments, we tabulated the notation in which responses to editorial comments (Q1) and substantive questions/remarks (Q2-Q26) were grouped.
|
Q1 |
Line 1: Change ‘Effect’ to ‘Effects’ |
|
Line 1: Change ‘montmorillonite’ to ‘montmorillonite and Gentamycin’ |
|
|
Line 13: Add ‘(PCL)’ after ‘poly-e-caprolactone’ |
|
|
Line 13: Change ‘2 %wt.’ to ‘2 wt.%’ |
|
|
Line 14-15: Change ‘pure polymer’ to neat ‘PCL’ |
|
|
Line 15: Remove ‘(PCL)’ |
|
|
Line 16: Remove ‘solvent system’ |
|
|
Line 18: Remove ‘preceding’ (it has a strike through it) |
|
|
Line 21: Change ‘MMT_G’ to ‘MMTG’ |
|
|
Line 38-39: Change ‘,such as, for example,’ to ‘such as’ |
|
|
Line 40: Change ‘goes back to’ to ‘is recycled into’ |
|
|
Line 43-44: Change ‘application possibilities’ to ‘possible applications’ |
|
|
Line 49: Define PCL and MMT |
|
|
Line 61: Change ‘(cotton matrix)’ to ‘using a cotton matrix’ |
|
|
Line 63: Change ‘difficult dispersion’ to ‘difficulty to disperse’ |
|
|
Line 65: Change ‘so much’ to ‘extensive’ |
|
|
Line 74: Change ‘achieving’ to ‘achieved’ |
|
|
Line 77: Remove the space after ‘bis-‘ |
|
|
Line 78-79: Rework this sentence to clarify the solvent method and drying |
|
|
Line 83: Change ‘application possibilities’ to ‘possible applications’ |
|
|
Q2 |
Line 86-88: Rework the sentence to clarify the electospinning process: the potential different between the nozzle and collector is the voltage; make this clear. The nozzle is charged with a high voltage, while the collector is grounded, this difference, causes the extraction of the polymer stream. It should also be included that the unstable whipping of this solution causes the polymer to stretch to obtain fibers with very small diameters and high surface to volume area. |
|
Q3 |
Line 86-88: The authors should include more than a single citation for the electrospinning process, as it is well studied. |
|
Q1 |
Line 89: Change ‘polymer volume’ to ‘polymer solution’ |
|
Line 92-93: Change ‘at the same time allows to achieve’ ‘simultaneously allows for’ |
|
|
Line 97: Change ‘ volume’ to ‘matrix’ |
|
|
Line 98: Change ‘ application potential’ to ‘potential applications’ |
|
|
Line 100: Change ‘the’ to ‘this’ |
|
|
Q1 |
Line 105: Clarify X-ray method: SEM, Powder, etc. |
|
Line 105: Change ‘particle distribution analysis’ to ‘dynamic light scattering’ |
|
|
Line 106: Change ‘pure’ to ‘neat’ |
|
|
Q4 |
Line 111: The authors need to define what ‘natural conditions’ are; there is no such thing. |
|
Q1 |
Line 116: State purity of MMT |
|
Line 117: Ensure the ® is superscripted 3 |
|
|
Line 119: State purity of gentamicin sulfate |
|
|
Line 120: Change ‘poli-e-kaprolakton’ to ‘poly-e-caprolactone’ |
|
|
Line 122: Change ‘pure’ to ‘neat’ |
|
|
Line 124: State whether or not everything was used as received |
|
|
Line 127-129: Rework sentence; it is not clear what was done: Example rework: Preparation of MMTG? began with calcination of montmorillonite at 200oC for 15 minutes. Then, it was intercalated with gentamicin sulphate in a water suspension at 50oC for 24 h |
|
|
Line 127-129: include if the synthesis was accomplished in a round bottom flask, or beaker? |
|
|
Line 127-129: The quantity in g or mol of the MMT and G needs to be included. |
|
|
Line 127-129: The authors may also want to cite their previous work |
|
|
Line 131: Change days to hours. |
|
|
Line 133: I believe the authors made MMTG here. They should include that in the first sentence of this paragraph instead of ‘starting ‘materials’ and remove the last sentence in this paragraph for clarity. |
|
|
Q5 |
Line 135: Rework sentence: …using a 4:3 v/v solution of DCM:DMF… |
|
Line 136: Clarify what mixture of solvents? Is it the DCM:DMF? |
|
|
Q1 |
Line 138: Change ‘a weight of polymer’ to ‘PCL’ |
|
Line 140: make sure ‘in’ is lower case |
|
|
Q6 |
Line 141: State temperature of water bath when using ultrasound |
|
Line 141: State manufacturer of ultrasound equiptment |
|
|
Q1 |
Line 144: Fix the reference to table 1 |
|
Line 145: Remove ‘Specification of’ |
|
|
Table 1: Ensure PCL_MMTG is on the same line of text 4 |
|
|
Q7 |
The authors must clarify why they inter-compare fibers using variables not discussed: the flow rate of stream; needle collect distance and voltage are all different |
|
Q8 |
Line 172: Can the authors clarify at what point the fiber mats were dried? In their calculation was the dried weight used before or after the incubation in water? |
|
Q1 |
Line 174: Change ‘formula’ to ‘equation’ |
|
Q9 |
Line 179: These are very long strips if 60 mm…. Can the authors clarify what ‘working area’ means? |
|
Q1 |
Line 185: Change ‘weigh’ to ‘weighed’ |
|
Line 186: Ensure the degree symbol is not underlined |
|
|
Line 189: The authors need to clarify what the variables are; it is confusing as is. Perhaps remove ‘-‘ and change it to ‘is the’ |
|
|
Check that all , used as decimals are changed to . there are several cases of this such as Line 195 |
|
|
Line 195: Change 37oC to 37 oC; remove the underscore |
|
|
Line 196: change days to h |
|
|
Line 200: change ‘formula’ to ‘equation’ |
|
|
Line 201-202: replace all of the ‘-‘ with ‘is the’ |
|
|
Q10 |
Line 203: Why would the authors press the fiber mats into pellets? They should’ve just been cut into square mats of a specific dimension and not pressed. By pressing the mats into pellets, this changes the morphology of the fibers, possible release kinetics of G, and potential structure of MMT or MMTG. |
|
Q1 |
Line 205: Change ‘5mm’ to ‘5 mm’ |
|
Line 209; Change ‘hours’ to ‘h’ |
|
|
Line 213: clarify ‘active substance’ |
|
|
Line 213: Fix the reference to Figure 1. |
|
|
Figure 1. Multiple x-ray data are offset on the same plot. Include a Y axis scale bar as an inset to give reads perspective. 5 |
|
|
Figure 1. X axis should have the theta symbol, not ‘theta’ |
|
|
Line 218-219: Can the authors justify 5-6 significant figures in their measurements? These data are usually reported with 3-4 significant figures. |
|
|
Line 222: Fix reference to Table 2 |
|
|
Table 2: Too many significant figures in the reported data. |
|
|
Line 229: clarify ‘active substance’ |
|
|
Line 231: Fix reference to Figure 2 |
|
|
Figure 2: Change ‘%vol.’ to ‘vol.%’ |
|
|
Figure 2. remove ‘(DLS method)’ |
|
|
Line 237-242: the incorrect symbol is used reporting the data; it should be: - not ÷ |
|
|
Q11 |
Figure 2: Can the authors comment on why the main peak for MMTG is not symmetrical? |
|
Q1 |
Line 247-248: Fix the reference to the figure |
|
Line 253: Clarify or remove ‘active substance’ |
|
|
Q12 |
Line 265-268: The authors need to state how the thickness was measured in the experimental. |
|
Line 265-268 The authors need to clarify why they used different electrospinning conditions to compared the fibers listed in Table 3. |
|
|
Line 269-271: The authors need to clarify the discrepancy in the thickness of membranes to the size of the fibers. I don’t see it, unless they aren’t including PCL in this statement. |
|
|
Q1 |
Line 271: Fix the reference to the Figure. |
|
Q12 |
Line 275-276: The average fiber sizes discussed here are different than is what’s listed in the table |
|
Q1 |
Line 278: Fix reference to the Figure |
|
Line 287: Change ‘PCL polymer’ to ‘neat PCL’ |
|
|
Line 293: Fix reference to Figure 6 |
|
|
Line 299: Fix reference to Figure |
|
|
Q13 |
Line 300: Authors need to explain why the wettability and water absorptivity of the membranes changes upon the addition of G. Does it change the morphology of the fibers? |
|
Q1 |
Figure 7: correct the Y axis to include the theta symbol |
|
Line 306: Fix the reference to the Table |
|
|
Line 313: Fix reference to Figure |
|
|
Q14 |
Line 320-322: The authors need to discuss the morphology of the fibers when MMT and MMTG are added rather than allude to possible changes – what are those changes? For Example, do the surfaces of the fibers with MMT and MMTG appear more heterogeneous via microscopy? I’d think that they do. |
|
Q15 |
Figure 8. Why did the authors cut off the data for PCL and PCL_G, but not the other fibers? |
|
Q16 |
Figure 8. It appears that there is some type of stain-hardening for PCL_MMTG. Can the authors comment on this? |
|
Q1 |
Line 328: Change ‘PCL alone’ to ‘neat PCL’ |
|
Line 329: Fix reference to Figure |
|
|
Line 332: Remove the underscore on the temperature symbol |
|
|
Q17 |
Line 336: Do the structural changes occur or no? This should be determined. Possibly by microscopy. The authors need to comment on the amount of MMT and MMTG in the fibers at this point. They should also discuss what the morphology is: does the location of the MMT and MMTG have a preference for the interior of the fibers? Or the surface? |
|
Q1 |
Figure 9: Change ‘egzo’ to ‘exo’ |
|
Line 339: Change ‘thermal study (DSC)’ to ‘DSC’ |
|
|
Line 347: Fix reference to Figure |
|
|
Line 349: Change ‘the inhibition zone was’ to ‘the zones of inhibition were’ |
|
|
Line 352: Fix reference to Table |
|
|
Q18 |
Table 5. Include the time at which the diameters were measured in the caption 7 |
|
Q1 |
Line 367-369: Change X days to X h. |
|
Q19 |
Line 369: Can the authors explain why intercalated G has greater antimicrobial efficacy than just G |
|
|
Figure 11. Change ‘czas’ to Time |
|
Q20 |
Figure 11. The table should be removed from the figure and be its own Table. Perhaps, remove the table in its entirety given the two graphs. |
|
Figure 11. the inset graph (left graph) should be to the right of the larger graph. How the graphs are displayed can be confusing. |
|
|
Q1
|
Line 378: Fix reference to Table |
|
Line 385: Fix reference to Table |
|
|
Line 387: Define ‘they’ |
|
|
Line 389: Fix reference to Table |
|
|
Line 397-399: Rework sentence; it is unclear/ |
|
|
Q21 |
Line 405: On what basis is the MMT membrane the most heterogeneous? There is not a discussion of the data to support this. |
|
Q1 |
Line 406: Fix reference to Figure |
|
Line 409: Fix reference to Figure |
|
|
Line 423: Fix reference to Figure |
|
|
Review the use of a comma (,) in place of period (.) for a decimal: Line 195, 219, 220,238, 239, Table 4, Figure 11 |
|
|
Q22 |
Line 424: Revise this sentence; A rotating collection will cause some alignment of the fibers, which is directly related to its rotational speed. A true ‘random’ orientation of the fibers will employ a stationary target such as a grounded aluminum plate. This is not what the authors did and should discuss. |
|
Q23 |
Line 429: The authors cannot assume these results are due to interfiber contacts. This is especially true because their MMT fibers have the larger fiber diameter. To make this statement, show the data to support this. Further, these ‘interfiber’ contacts are usually a result of poor electrospinning parameters: too short of 8 working distance, relative humidity during spinning, too slow rotational speed of collector, etc. See work of Jan Lagerwall. |
|
Q24 |
Line 431-435. I do not agree that this has to do with solvation of the MMT. This is related to the average fiber diameter and a lack of optimal spinning conditions (see above). |
|
Q1 |
Line 439: Fix reference to Figure |
|
Line 451-455: Fix references to Figures |
|
|
Q25 |
Line 461: Rework… ‘Probably the kinetics of gentamicin’… The authors cannot make an assumption here. Are the kinetics related to wettability and porosity? Fiber diameter? Location of MMTG within the fiber? The authors need to add a Figure on this data to discuss. Perhaps, plot wettability and/or porosity versus the release time at a specific time for each fiber. Why does G have an increase in antimicrobial efficacy when intercalated with MMT? Based upon their data the answer will be most easily explained on the morphological changes to the fibers. |
|
Q26 |
Line 482: MMT_G is not reference yet, nor is it a membrane I belive. Do the authors mean something else? |
Q2-Q3. Rework the sentence to clarify the electospinning process: the potential different between the nozzle and collector is the voltage; make this clear. The nozzle is charged with a high voltage, while the collector is grounded, this difference, causes the extraction of the polymer stream. It should also be included that the unstable whipping of this solution causes the polymer to stretch to obtain fibers with very small diameters and high surface to volume area
The idea of the electrospinning process has been in use since 1950 and therefore its fundamentals are well known to those working on electrospun materials. Detailed data on the fundamentals of the process can be found in numerous review papers. Our work is experimental in nature and shows one of the many possibilities of this process. What remains unchanged is the fact that the conditions of the electrospinning process are fine-tuned each time: due to the capabilities of the apparatus, the polymer and solvent system used as well as the environmental conditions.
As it is known, taking into account three groups of parameters, which are to guarantee the effective course of electrospinning, is a compromise between the conditions in the solution, the conditions of the process and the environment in which the process takes place. The electrospinning process takes advantage of the potential difference between the earthed collector and the tip of the nozzle, which creates a strong electric field. In our case, an additional disturbance of the solution was the presence of insulating ceramic nanoparticles, which made it necessary to increase the voltage from 15 to 18 kV and the speed of feeding the polymer with the modifier to the nozzle of the device. Leaving the polymer to low flow caused clogging of the nozzle or the appearance of defects in the form of polymer droplets (beads) on the manifold. On the other hand, too high feeding speed at the assumed needle-collector distance (typically 10 cm for PCL) caused sticking of the polymer fibers, as a consequence of which a polymer film was formed instead of the morphology of the fibers. Ultimately, the remoteness of the nozzle and the use of temperature and humidity control with the proposed solvent system (volatile DCM and less volatile DMF) allowed for the production of good quality fibers.
The polymer solution used to produce nanofibers must have fiber-forming properties, i.e., sufficiently high viscosity and electrical conductivity [1-6]. The viscosity of the solution is closely related to the molecular weight of the polymer and the amount of polymer in the solution-the concentration of the solution, which plays a key role in the fiber formation process [2]. In order to preserve the fiber-forming properties of the solution, the polymer, in addition to its high molecular weight, should be characterized by a linear chain structure without the presence of side chains (hence, linear PCL was proposed in our work). The presence of polar groups in the polymer chain allowing its dissolution is also important. The solvents used should be volatile (this role was played by DCM) and should increase the dielectric properties of the polymer solution (this role was played by DMF). It is also important that the solvent has a low surface tension (e.g. DMF, dielectric constant at 20 ºC - 37.06 [1], DCM, high volatility, surface tension 27.2 mN/M at 25 ºC [5]). The viscosity of the polymer solution determines the final product of electrospinning; as the viscosity increases, up to its optimally high value, homogeneous nanofibers are obtained without "beaded" thickening, occurring at insufficiently high viscosity of the polymer solution [6]. It is worth mentioning, however, that too high viscosity or the presence of fillers may cause difficulties with the ejection of the polymer bundle from the needle lumen and its clogging, so it is important to maneuver the concentration of the solution and the set of the solvent system.
The speed at which the polymer solution is fed through the infusion pump and the distance between the end of the needle and the collector, among other things, are selected in terms of the time required for polymer solution charge polarization and complete evaporation of the solution solvent [6]. Too low a voltage can cause problems with pulling the polymer out of the needle lumen (problem with overcoming the surface tension of the polymer solution). The conditions and composition of the atmosphere prevailing during electrospinning, the so-called ambient conditions, affect the stability of the fiber manufacturing process [7]. Only under certain environmental conditions - an appropriate range of temperature and humidity, the polymer has a chance to take the form of a stable jet in an electric field. Low humidity and high temperature contribute to faster evaporation of the polymer solution solvent by reducing its viscosity [8].
[1] Xianrui Xie et al. Electrospinning nanofiber scaffolds for soft and hard tissue regeneration, Journal of Materials Science & Technology, t. 59, s. 243–261, 2020, doi: 10.1016/j.jmst.2020.04.037.
[2] Hassan M. Ibrahim et al. A review on electrospun polymeric nanofibers: Production parameters and potential applications, Polymer Testing, t. 90, s. 106647, paź. 2020, doi: 10.1016/j.polymertesting.2020.106647.
[3] A. Keirouzet al. 2D and 3D electrospinning technologies for the fabrication of nanofibrous scaffolds for skin tissue engineering: A review, WIREs Nanomedicine and Nanobiotechnology, t. 12, nr 4, s. e1626, 2020, doi: 10.1002/wnan.1626.
[4] J. Wu et al. Enhancing cell infiltration of electrospun fibrous scaffolds in tissue regeneration, Bioactive Materials, t. 1, nr 1, s. 56–64, 2016, doi: 10.1016/j.bioactmat.2016.07.001.
[5] Mirjalili, M et al., Review for Application of Electrospinning and Electrospun Nanofibers Technology in Textile Industry. J. Nanostruct Chem. 2016, 6, 207–213.
[6] Ghosal, K. et al., I. Electrospinning Over Solvent Casting: Tuning of Mechanical Properties of Membranes. Sci. Rep. 2018, 8, 5058
[7] Persano, L. et al. Industrial Upscaling of Electrospinning and Applications of Polymer
Nanofibers: A Review. Macromol. Mater. Eng. 2013, 298, 504–520.
[8] J. Rnjak-Kovacina et al. Increasing the Pore Size of Electrospun Scaffolds, Tissue engineering. Part B, Reviews, t. 17, s. 365–72, 2011, doi: 10.1089/ten.teb.2011.0235.
Q4. The authors need to define what ‘natural conditions’ are; there is no such thing.
Thank you for your comment; we used the wrong term to describe the process of biodegradation which is a process that occurs in a natural environment and is the result of a synergistic effect on the material by microorganisms, water and the ions it contains.
Biodegradation is a natural process by which organic chemicals in the environment are converted to simpler compounds, mineralised and redistributed through elemental cycles such as the carbon, nitrogen and sulphur cycles [1]. Biodegradation can only occur within the biosphere as microorganisms play a central role in the biodegradation process. There are four biodegradation environments for polymers and plastic products: soil, aquatic, landfill and compost [2]. Each environment contains different microorganisms and has different conditions for degradation. Of course the biodegradation rate in bionanocomposites depends on a number of factors including fibre content, the biodegradability of each component and the quality of the interface [3].
[1] A. Hodzic, Green Composites, Woodhead Publishing Series in Composites Science and Engineering 2004, Pages 252-271
[2] D.Plackett, A.Vázquez, Natural polymer sources, Polymer Composites and the Environment 2006, Pages 123-153
[3] Ali Chamas, Hyunjin Moon et al. Degradation Rates of Plastics in the Environment, ACS Sustainable Chem. Eng. 2020, 8, 9, 3494–3511
Q5. Rework sentence: using a 4:3 v/v solution of DCM:DMF. Clarify what mixture of solvents? Is it the DCM:DMF?
As mentioned in the description of the article's methodology, in our study we used a mixture of DCM (dichloromethane): DMF (dimethylphoramide) solvents in a volume ratio of 4:3, which were used to dissolve the PCL pellet - to prepare a 15% fiber solution used for the electrospinning process. No other solvents were used in our study.
Q6. State temperature of water bath when using ultrasound. State manufacturer of ultrasound equiptment
Thank you for your attention. Homogenization of the nanofiller was carried out in a water bath at room temperature with the Sonix VCX 130PB apparatus. No changes of the bath temperature were recorded as this was not the purpose of the study - only the homogenization of the polymer-filler suspension. The purpose of the applied cooling conditions (water bath) was to prevent the thermal degradation of the polymer.
Q7. The authors must clarify why they inter-compare fibers using variables not discussed: the flow rate of stream; needle collect distance and voltage are all different
Thank you for comment. Missing data have been completed and highlighted in color in the manuscript
Q8. Can the authors clarify at what point the fiber mats were dried? In their calculation was the dried weight used before or after the incubation in water?
Thank you for your comment. We used the traditional test of mass absorption, defined as the ability of a material to absorb water at the maximum water saturation of the material. We used the formula known from the literature, which clearly defines the meaning of the method as: the ratio of the mass of absorbed water to the dry mass of the sample.
After incubation time, the membranes were reweighed (the wet membrane mass) then drying them, and the obtained results of the dry sample mass (mp, g). This information were used to determine the water absorption (N).
Q9. Can the authors clarify what ‘working area’ means
The working area is the distance between the jaws of the testing machine where the material is placed, or the area of material that is being tensile tested.
Q10. Why would the authors press the fiber mats into pellets? They should’ve just been cut into square mats of a specific dimension and not pressed. By pressing the mats into pellets, this changes the morphology of the fibers, possible release kinetics of G, and potential structure of MMT or MMTG
Thank you for your kind comment. Of course, the fibrous membranes were cut to 5 mm diameter circles, we also tested gentamicin intercalated powders and they were cut to 5 mm diameter pellets - that is why there was a mistake in preparing the samples for microbiological tests.
Q11. Figure 2: Can the authors comment on why the main peak for MMTG is not symmetrical?
Since there are crystallographic planes that are not parallel to the surface of the crystal, the angle of incidence of the beam on the sample surface may differ significantly from the Bragg angle for a given family of planes. Perhaps we are dealing with such a case in the grain of the natural clay that is MMT
Q12. The authors need to state how the thickness was measured in the experimental. The authors need to clarify why they used different electrospinning conditions to compared the fibers listed in Table 3. The authors need to clarify the discrepancy in the thickness of membranes to the size of the fibers. I don’t see it, unless they aren’t including PCL in this statement.
The thickness of the fibrous membranes was determined under a microscope (Figure 1). The cross-section of the membrane was photographed and measured using microscope software. Five to eight measurements were made on one cross-section, and the mean value and standard deviation were determined on the basis of the obtained results.
Missing data on the electrodeposition and the influence of process conditions on the obtained fibers were completed in the manuscript.
As is well known, consideration of the three groups of parameters to ensure successful spinning is a compromise between solution conditions, process conditions, and the environment in which it occurs. The electrospinning process is the exploitation of the potential difference between the grounded collector and the nozzle end, which results in the generation of a strong electric field. In our case, an additional disturbance to the solution that the electric field had to deal with is the presence of ceramic nanoparticles (insulators). It was therefore necessary to increase the voltage from 15 to 18kV and the speed of feeding the polymer with modifier to the nozzle of the device. Leaving a low polymer flow rate caused the nozzle to pivot or the appearance of defects in the form of polymer droplets (beads Figure 1a) or thick polymer plumes deposited on the collector (Fgura 1b). On the other hand, too high delivery speed at the assumed needle-collector distance (typical for PCL 10cm) causes polymer fibers to stick together, resulting in polymer film instead of fiber morphology (Fig. 1d). Moving the nozzle away and using temperature and humidity control with the proposed solvent system (volatile DCM and less volatile DMF) enabled the formation of a nonwoven film (Figure 1c). The developed conditions were optimized until a satisfactory morphology of fibers included in the manuscript was obtained. Data of the process is given in Table 1 in the manuscript.
Figure 2. Defects on trawls appearing during optimization process of electrowinning of trawls based on PCL_MMT and PCL_MMTG: beads (a), polymer spitting (b), fiber merging (c,d).
Q13. Authors need to explain why the wettability and water absorptivity of the membranes changes upon the addition of G. Does it change the morphology of the fibers?
The high wetting angle value is due to the rough microstructure of the membrane resulting from the presence of fibers. Wetting of the material in such a form is difficult due to the higher surface expansion and longer time necessary for water to penetrate the irregular material surface [1]. Therefore, the wetting angle for PCL is the highest. The phenomenon which occurs during wetting of submicron and nanometric surfaces is known in literature as Cassie's wetting model [2].
The statement above is supported by the literature. Sołtysiak et al. show that even cast PCL films into which MMT particles were introduced gain in roughness - the presence of MMT particle influences the ease of polymer spherulites formation and MMT itself can act as a factor initiating nucleation of polymer chain. In the discussed case of fibrous material, we can observe the effect of increase in the fiber diameter due to the presence of MMT filler in its volume (Figure 4 in the manuscript). An increase in the fibre diameter and a wider fibre size distribution is an effect of 'smoothing' of the nonwoven surface, and thus a better and faster penetration of the surface by water droplets (easier dispersion), which results in a smaller value of the wetting angle. Further decrease in contact angle can be caused by the presence of gentamicin not only between montmorillonite layers but also by the presence of particles of genramycin sulfate salt on the surface of powder added to electrospinning. This assumption is confirmed by almost identical value of contact angle for PLA_G and PLA_MMTG materials.
[1] Anish Tuteja, Wonjae Choi et al. Design parameters for superhydrophobicity and superoleophobicity, MRS Bulletin 2008 33 (8), 752-758
[2] D. Zhang, L. Wang, H. Qian, et al. Superhydrophobic surfaces for corrosion protection: a review of recent progresses and future directions. J Coat Technol Res 13, 11–29 (2016)
[3] E. Sołtysiak, M. Błażewicz, influence of nanoparticles on physical and chemical surface material properties, Eng Biomat, 92, (2010), 30-3
Q14. The authors need to discuss the morphology of the fibers when MMT and MMTG are added rather than allude to possible changes – what are those changes? For Example, do the surfaces of the fibers with MMT and MMTG appear more heterogeneous via microscopy? I’d think that they do.
Thank you for your comment, however, we have not observed any influence of nanofiller on the morphology of the fibers; frankly speaking, MMT was hidden inside the fibers. We did not find any changes in the fiber morphology due to humidity and phase inversion (exchange of DMF into water and appearance of pores on the fiber surface). Thus, there is no mention in the manuscript on this point.
Q15. Why did the authors cut off the data for PCL and PCL_G, but not the other fibers
Thank you for your attention, we did not truncate the data for either PCL or PCL_G, the tests were performed on 5-8 samples and the material with repeated mechanical characteristics was selected for the summary presented in the graph.
Q16. It appears that there is some type of stain-hardening for PCL_MMTG. Can the authors comment on this.
Thank you for you comment. In man publication these phenomena was explain [1-3]. In analogy to rubber elasticity at large deformations, strain hardening in polymer composites is typically attributed to the increasing resistance to deformation of extended and oriented polymer chains (2–4). However, it has been shown that polymer chain alignment during strain hardening is strongly affected by dispersing fillers within the host polymer matrix (5–6).
Probably we have this effect into our fibrous nanocomposite. To account for these observations, one needs to establish the relation between the macroscopically observed strain hardening and the microscopic chain alignment that is affected by the presence of fillers. The connection between chain alignment and strain hardening in glassy polymer composites is purported to occur because the nanofillers act as “entanglement attractors. In this picture, the segmental mobility of the polymer is disturbed (e.g., strongly constrained) by the presence of a large amount of surface area of the nanofillers, causing an increase in the number of physical entanglements; this results in greater alignment of effectively shorter segments between entanglement points in response to the applied load.
Unfortunately, we did not conduct such detailed studies on the mechanical fusion of the results with the filler dispersion - this is certainly a very interesting direction of research and we will try to develop it in the future
[1] Fried JR (2003) Polymer Science and Technology (Prentice Hall, Englewood Cliffs, NJ).
[2] Riande E, Diaz-Calledja R, Prolongo MG, Masegosa RM, Salom C (2000) Polymer Viscoelasticity: Stress and Strain in Practice (Marcel Dekker, New York)
[3] Haward RN (1993) Strain hardening of thermoplastics. Macromolecules 26:5860–5869
[4] Na B, et al. (2007) Inverse temperature dependence of strain hardening in ultrahigh molecular weight polyethylene: Role of lamellar coupling and entanglement density J Phys Chem B 111:13206–13210.
[5] Jancar J, Hoy RS, Lesser AJ, Jancarova E, Zidek J (2013) Effect of particle size, temperature, and deformation rate on the plastic flow and strain hardening response of PMMA composites. Macromolecules 46:9409–9426.
[6] Pukanszky B (1990) Influence of interface interaction on the ultimate tensile properties of polymer composites. Composites 21:255–262
Q17. Do the structural changes occur or no? This should be determined. Possibly by microscopy. The authors need to comment on the amount of MMT and MMTG in the fibers at this point. They should also discuss what the morphology is: does the location of the MMT and MMTG have a preference for the interior of the fibers? Or the surface?
We have already explained in Q14 ze on none of the fabricated nanocomposite fibers: PCL_MMT, PCL_G or PCL_MMTG we did not find particles of aluminosilicate or gentamicin sierate outside the fibers. Thus, it seems that these particles are comapeptive to the prepared polymer solution and locate inside the fibers during spinning. This is confirmed by direct evidence; SEM images, but also by indirect evidence, i.e. mechanical and thermal tests; the increase in strength relative to pure CPL as well as the increase in crystallinity proves that we have succeeded in introducing the nanofiller into the fiber.
Q18. Include the time at which the diameters were measured in the caption 7
Microbiological tests were performed according to the protocol of the American Society for Microbiology. The zones of inhibition were read between 20-22h after inoculation of bacteria on agar [1].
[1] J. Hudzicki, Kirby-Bauer Disk Diffusion Susceptibility Test Protocol, American Society for Microbiology © 2016
Q19. Can the authors explain why intercalated G has greater antimicrobial efficacy than just G
Our observations show that despite repeated washing out of MMT after the inetercalation process (MMTG), a small part of gentamicin remains on the surface or on the edges of MMT flakes. This phenomenon should not be surprising; there are numerous examples in the literature of various agents that do not intercalate the interlaminar spaces (e.g. Mg2+) but adsorb on the edges of planes and thus have an easier route in migration out of the fiber during the incubation process.
In montmorillonite crystal structure, aluminum ions in aluminum-oxygen octahedron are easily replaced by various ions such as magnesium or iron, while the silicon ions in silicon-oxygen are easily replaced by aluminum. Meantime, cations like Na+ or Ca2+ can be filled into the interlayer space to balance the charge, ascribed to the weak interaction force, such as Van de Waals force, between them. Cations in the interlayer of MMT, such as Na+ and Ca2+, are hydrated, so the spacing of MMT is actually a function of water content. With different types of interlayer cations, the distribution, hydration, and diffusion properties in the interlayer of MMT are different [1-3]. The process of organic cations intercalating MMT is the interaction of multiple forces, the research and description on intercalation process and mechanism are not clear. The researchers did not focus on interaction force between layer and interlayer cations [4].
[1] Jong Sun Jung et al. Application of smectite for textile dyeing and fastness improvement, RSC Advance, Issue 63, 2019
[2] H. C. Greenwell, L. A. Bindley, P. R. Unwin et al., In situ monitoring of crystal growth and dissolution of oriented layered double-hydroxide crystals immobilized on silicon, Journal of Crystal Growth, vol. 294, no. 1, pp. 53–59, 2006.
[3] P. P. Kumar, A. G. Kalinichev, and R. J. Kirkpatrick, Hydration, swelling, interlayer structure, and hydrogen bonding in organolayered double hydroxides: insights from molecular dynamics simulation of citrate-intercalated hydrotalcite, Journal of Physical Chemistry B, vol. 110, no. 9, pp. 3841–3844, 2006.
[4] V. Aggarwal, Y. Y. Chien, and B. J. Teppen, Molecular simulations to estimate thermodynamics for adsorption of polar organic solutes to montmorillonite, European Journal of Soil Science, vol. 58, no. 4, pp. 945–957, 2007.
Q20. The table should be removed from the figure and be its own Table. Perhaps, remove the table in its entirety given the two graphs. The inset graph (left graph) should be to the right of the larger graph. How the graphs are displayed can be confusing
Thank you for your comment, rearranging the data does not change its value.
Q21. On what basis is the MMT membrane the most heterogeneous?
Thank you for your comment, this wording is incorrect and has been removed
Q22. Revise this sentence; A rotating collection will cause some alignment of the fibers, which is directly related to its rotational speed. A true ‘random’ orientation of the fibers will employ a stationary target such as a grounded aluminum plate.
Thank you for your comment this sentence was corrected and replace into manuscript
Q23. The authors cannot assume these results are due to interfiber contacts. This is especially true because their MMT fibers have the larger fiber diameter. To make this statement, show the data to support this. Further, these ‘interfiber’ contacts are usually a result of poor electrospinning parameters: too short of 8 working distance, relative humidity during spinning, too slow rotational speed of collector, etc. See work of Jan Lagerwall
Thank you for your comment, it is an important remark. The analysis of changes occurring in the force-strain system and microscopic observation of nonwovens, as well as subsequent microscopic observations (SEM) of materials indicate that our fibers nevertheless have contact points described in the literature as interfiber (Figure 1). Obviously, it can be related both to underdeveloped electrospinning conditions, but also can be an effect of the presence of a specific nanofiller in the form of layered aluminosilicate.
Our unpublished data that we wish to present here to support this thesis concerns structural studies of MMT, MMTG and G powders. The FTIR-DRIFT technique is used to characterize the surface of the particles. The spectrum presented below shows that although the bands of gentamicin sulfate overlap in many places with those of MMT, it is possible to distinguish those frequency ranges that belong to gentamicin Figure 2). This means that there is an insignificant amount of salt on the plate surface/edge of the MMT. In addition, the wettability and absorptivity of these powders also change (tests performed for MMT and MMTG using Sigma 700/701 force tensiometers according to the Washburn theory). The results obtained show that while MMT is hydrophilic in nature (contact angle of 68.9±4.3), modification of the clay with gentamicin sulfate causes a decrease of the angle by nearly 5 degrees (to 62.2±3.5). This behavior confirms that our MMT powder undergoing modification changes not only microstructurally (SEM observations included in the manuscript) but also structurally. This change can influence the wettability of the filler during homogenization with polymer solution as well as during electrospinning.
Figure 1. Morphology of CL-based fibrils with MMT (a) and MMTG (b) additions, ageing indicates areas which according to the authors correspond to the literature description of interfiber contact.
Figure 2. Compilation of the FTIR-DRIFT spectrum showing significant differences in band positions that are due to surface changes resulting from the modification of MMT by gentamicin sulfate.
By complementing our observations with literature data from the source indicated by the reviewer [1], we can conclude that although the correctness of the electrospinning process depends largely on the humidity in the device chamber during the formation of the Taylor cone, in this particular case the solvent system used is not very polar compared to those indicated in the manuscript of Prof. Lagerwall from 2020 (there they worked on ethanol and PVP). Nevertheless, note that DMF is a solvent that readily combines with water. However, its amount (flake surface with electrostatically bound sulfate cations) is minimal which, according to us, should not significantly affect the electrospinning process. We did not find any inaccuracies in the formation of the Taylor cone during the process (humidity in our chamber fluctuated within the range of RH 20-25%), although we cannot exclude with certainty the situation of disturbing the electrospinning process by the process described in the item indicated. This matter will certainly be further investigated by us - thank you for the tip.
[1] Catherine G. Reyes and Jan P. F. Lagerwall, Disruption of Electrospinning due to Water Condensation into the Taylor Cone, ACS Appl. Mater. Interfaces 2020, 12, 26566−26576
Q24. I do not agree that this has to do with solvation of the MMT. This is related to the average fiber diameter and a lack of optimal spinning conditions (see above).
We agree with the reviewer's opinion; we used the wrong word; it was not the solvation but the wettability of the powder through the solvent system that was crucial. The higher absorption of DCM:DMF by the powders and the more difficult evaporation can be the reason for the interfiber formation (which affects the mechanical properties). On the other hand, the increase in fiber diameter can be largely attributed to imperfect homogenization of the aluminosilicate nanofiller. Such behavior in literature is also observed in fibers in which halloysite tubes were used as a filler [1]. In this case as well as in our study, no changes can be seen within the fiber morphology and fiber surface topography; there is no surface wasting due to filler exposure (see above on Figure 1).
The data published by Hao Yi aet al. show that the MMT surface interacts weakly with water molecules, whereas exchangeable cations adsorbed on the MMT surface have a very strong affinity for water molecules - hence our term - solvation.
In fact, exchangeable cations are electrostatically adsorbed on the MMT surface in the form of a hydration shell with a cation core and an envelope of water molecules [2]. In our case, if there are sulfate ions on the flake surface then they are surrounded by water, and more importantly this is the property that plays a dominant role in giving the hydrophilic character to the MMT surface. The higher number of exchangeable cations and stronger hydration capacity of cations on the surface of MMT plays a dominant role in making it hydrophilic. Obviously, in our case we do not have water in the system as the solvent system used for spinning is DCM:DMF but as mentioned above DMF readily combines with water which may influence the undeveloped electrospinning conditions resulting in fiber bonding. The unfortunate wording regarding solvation has been removed from the manuscript.
[1] Ganesh Nitya et al. In vitro evaluation of electrospun PCL/nanoclay composite scaffold for bone tissue engineering, Journal Materials Science: Materials in Medicine 23 (2012) 1749-1761
[2] Hao Yi et al. Surface wettability of montmorillonite (0 0 1) surface as affected by surface charge and exchangeable cations: A molecular dynamic study, Applied Surface Science 459 (2018) 148-154
Q25. Probably the kinetics of gentamicin’… The authors cannot make an assumption here. Are the kinetics related to wettability and porosity? Fiber diameter? Location of MMTG within the fiber? The authors need to add a Figure on this data to discuss. Perhaps, plot wettability and/or porosity versus the release time at a specific time for each fiber. Why does G have an increase in antimicrobial efficacy when intercalated with MMT? Based upon their data the answer will be most easily explained on the morphological changes to the fibers.
MMT_G is not reference yet, nor is it a membrane I belive. Do the authors mean something else
Thank you for your comment. Our assumption, which we included in the manuscript, is based on previous results already published concerning the powder materials themselves. There, gentamicin was released faster if it was bound to MMT (the intercalation method was also analogous) even if a different polymer matrix and a different method of nanocomposite formation were used. Additionally, our suspicions may be strengthened by two earlier answers to the reviewer's questions: on the surface of MMTG we have a small amount of sulfate itself, different powder morphology and better wettability, thus it can be confirmed that this is the reason for the different kinetics of release of the active compound from the fiber. Unfortunately, we did not observe the fibers after the release process, but it seems that this time of 12 days is too short to observe significant changes in the morphology of the PCL-based fiber. Information on the release kinetics and explanation of the changes occurring during the process was already required by another reviewer, so it has been expanded and completed in the manuscript.

Reviewer 3 Report
The paper entitled “Effect of montmorillonite addition on the properties of electrospun polycaprolactone fibers” by Ewa Stodolak-Zych, Roksana Kurpanik, Ewa Dzierzkowska, Marcin Gajek, Łukasz Zych, Karol Gryń and Alicja Rapacz-Kmita, presents the preparation and characterization of electrospun polycaprolactone fibers loaded with montmorillonite, montmorillonite intercalated with gentamicin sulfate and gentamicin sulfate. There are few reports in the scientific literature which describe the preparation and characterization of polycaprolactone fibers loaded with montmorillonite or other blends, however none of them treated the above combination. Natheless, some minor lacks can also be find in this paper, as follows:
- The Abstract should be improved and made more attractive.
- A lot of typing errors should be corrected (e.g. sulfate vs. sulphate)
- The authors should specify the chamber temperature of electrospinning setup, which is also an important fiber preparation parameter.
- Why the 2θ range was set as 5-13˚ considering that the montmorillonite also presents peaks at other 2θ values?
- Are the fiber diameters of the order of μm or nm? In text, the diameters are expressed as μm and in Figure 6 are nm.
- An inset with SEM images at high magnification should be added in order to see the difference between the fibers.
- No important differences between the DCS analyzes were observed, so maybe the Figure 9 should be removed from manuscript.
- The protocol for bacteria growth should be mentioned at characterization section.
- The title could be improved.
Author Response
Dear Reviewer,
Thank you for careful reading our manuscript entitled: Effects of montmorillonite and gentamicin addition on the properties of electrospun polycaprolactone fibers (Materials-1361097). We hope that the answers below will provide a better explanation of our manuscript.
Q1. The Abstract should be improved and made more attractive.
Abstract has been corrected and completed with the data requested by the reviewer. We hope that in its present form the abstract is more attractive for Rewiver. Corrected and highlighted in yellow in the text.
Q2. A lot of typing errors should be corrected (e.g. sulfate vs. sulphate)
Thank you for your attention, we have unified the name of the salt; both sulpahe and sulfate are correct names, but in the manuscript only gentamicin sulfate was left.
Q3. The authors should specify the chamber temperature of electrospinning setup, which is also an important fiber preparation parameter.
The temperature information was completed in Table 1. This was omitted by us as we tried to maintain similar ambient conditions (temperature, humidity) during electrospinning developed for pure fibers from PCL (21oC, 20-25%).
Only under certain environmental conditions - the right range of temperature and humidity, the polymer has a chance to take the form of a stable jet in an electric field. Low humidity and high temperature contribute to faster evaporation of the polymer solution solvent by reducing its viscosity.
The solvent system used, consisting of DCM (dichloromethane) and DMF (dimethylformamide), is characterized by low vapor pressure. Higher lethality is exhibited by DCM while DMF plays the role of a factor increasing the dielectric properties of the polymer solution. Too high a chamber temperature during electrospinning can cause clogging or reduce the efficiency of the process by pinching the device nozzles. Too high humidity can lead to phase inversion between DMF and water (from steam). As a result, it is difficult to obtain fibers with a smooth surface and often even the fiber form itself
Q4. Why the 2θ range was set as 5-13˚ considering that the montmorillonite also presents peaks at other 2θ values?
Thank you for your question. The degree of intercalation of this mineral can only be assessed on the basis of the position of the main peak d001 of montmorillonite. The change in the position of the d001 MMT peak, and more specifically its shift towards lower values of the 2θ angle, informs about the enlargement of the MMT interlayer space, which indicates the intercalation process and the incorporation of the active substance between the galleries of the mineral. This phenomenon is known from the literature on the subject [1]. The remainder of the MMT X-ray diffraction pattern after intercalation usually remains the same.
[1] Kamal Yusoh et al. Surface Modification of Nanoclay for the Synthesis of Polycaprolactone (PCL) – Clay Nanocomposite, https://doi.org/10.1051/matecconf/201815002005
Q5. Are the fiber diameters of the order of μm or nm? In text, the diameters are expressed as μm and in Figure 6 are nm.
Fibre diameters are micrometric (or more precisely submicrometric; less than 1um). In the literature on electrospinning when showing the distribution of fibre sizes usually nanometres are used even though the fibre diameters tend to oscillate around submicron but this allows for a clearer form of the graph: the fibre diameters then have integer values. This avoids the fractional values that make reading difficult.
Q6. An inset with SEM images at high magnification should be added in order to see the difference between the fibers.
Thank you for your attention. Both the magnification and the scaling bar are from the original images in the manuscript (ie Figure 3)
Q7. No important differences between the DCS analyzes were observed, so maybe the Figure 9 should be removed from manuscript.
Thank you for your comment. Unfortunately, we disagree with the reviewer's statement about the irrelevance of the data resulting from DSC. In the case of polymeric materials, it is important to perform thermal studies with special emphasis on DSC. This is important, especially when the polymer matrix is enriched with nanofiller. The nanoparticle can act as a nucleating agent, facilitate the formation of spherulites as shown by studies [1]. In case of polymers, we more often use DSC technique than XRD technique, where they are mostly amorphous group (less often semicrystalline). In the case of PCL, which is a semi-crystalline polymer, the filler acts just like that; the crystallinity of the material changes, which can be quantified from the DSC curves. Such results are given in Table 7. Performing XRD in the studied PCL_MMT system and its similar materials is difficult due to the thickness of the fibrous membrane itself (180-230um), the amount of filler introduced (only 2 %wt/wt) and the XRD beam itself would give a less conclusive result than a DSC test giving results with 3.5mg of nonwoven fabric.
[1] E. Sołtysiak, M. Błażewicz, influence of nanoparticles on physical and chemical surface material properties, Eng Biomat, 92, (2010), 30-3
Q8. The protocol for bacteria growth should be mentioned at characterization section.
Thank you for your comment. A reference to the protocol [1] according to which the microbiological tests were performed has been included in the literature section. It is referenced in the microbiological testing methodology in the text of the manuscript.
[1] J. Hudzicki, Kirby-Bauer Disk Diffusion Susceptibility Test Protocol, American Society for Microbiology © 2016
Q9. The title could be improved.
The title of the manuscript has been corrected
Reviewer 4 Report
The research articles entitled “Effect of montmorillonite addition on the properties of electrospun polycaprolactone fibers” by Ewa Stodolak-Zych et al.. In this study authors have developed the nanocomposite electrosun fibers of polycaprolactone (PCL), and in mixed with montmorillonite (MMT) together with antibacterial gentamicin (G) with improved mechanical properties for antibacterial application. Moreover, they have shown that electrospinning techniques is very effective for nanocomposite fibers formation of non-intercalated and intercalated montmorillonites. I will recommend this articles for publication after major revision.
- Abstract should be thoroughly revised. Some minor change:
- Page 1, line 13, “polycaprolactone (PCL)”,
- Page 1, line 15; “….as reference material (PCL)“ should be changed to “…..as reference material PCL”.
- Page 1, line 20, authors should write full name of “EDS”.
- Page 1, line 21, “MMT_G” should be changed to “MMTG”. Authors should consistence with acronyms used in the manuscript.
- Although, authors have written details introduction and explain the nanocomposite materials, antibacterial nanoparticles, MMT, gentamicin, however introduction lack the information of PCL and it’s biomedical role. Therefore, I will recommend authors to add some basic information and biomedical role of PCL in the introduction section. Following references can be suitable Polymers 2018, 10, 1387; doi:10.3390/polym10121387, Acta Biomaterialia 108 (2020) 97–110, and BioMed Eng OnLine (2017) 16:40.
- In materials Methods section authors should clearly provide the details of PCL and MMT amount (in wt%) in electrospinning process. Page 3, line 140 authors calculate 2 wt.% amount of filler, which is also mentioned in the Abstract (page 1 line 13). What is the amount of PCL ? Pls also write the details of gentamicin amount. Please mentioned it.
- Figure 1, Why gentamicin does not exhibit any peak in XRD analysis? What does it mean?
- PCL is a hydrophobic material. Figure 7, why addition of MMT, G, and MMTG in PCL reduce the water contact angle? Authors should provide bar graph.
- Figure 8, authors should provide tensile stress vs tensile strain curve as explain in (Fig.4 of ref. ACS Appl. Mater. Interfaces 2015, 7, 8088−8098). Why the tensile strain for nanocomposite nanofibers is higher than the pure PCL nanofibers? The data for PCL-MMTG and PCL-MMT showing fluctuations, why? It would be good if authors can provide smooth curve for these two materials.
- 11. The release of gentamicin showing burst release at early time point and exhibits long-term release up to 12 days why? Please explain what mechanism involve in the release process?
Author Response
Dear Reviewer,
Thank you for careful reading our manuscript entitled: Effects of montmorillonite and gentamicin addition on the properties of electrospun polycaprolactone fibers (Materials-1361097). We hope that the answers below will provide a better explanation of our manuscript.
Q1. Abstract should be thoroughly revised. Some minor change:
Abstract has been corrected and completed with the data requested by the reviewer. We hope that in its present form the abstract is more attractive for Rewiver.
Q2. Page 1, line 13, “polycaprolactone (PCL)”, Page 1, line 15; “….as reference material (PCL)“ should be changed to “…..as reference material PCL”. Page 1, line 20, authors should write full name of “EDS”, Page 1, line 21, “MMT_G” should be changed to “MMTG”. Authors should consistence with acronyms used in the manuscript.
Corrected and highlighted in yellow in the text.
Q3. Although, authors have written details introduction and explain the nanocomposite materials, antibacterial nanoparticles, MMT, gentamicin, however introduction lack the information of PCL and it’s biomedical role. Therefore, I will recommend authors to add some basic information and biomedical role of PCL in the introduction section. Following references can be suitable Polymers 2018, 10, 1387; doi:10.3390/polym10121387, Acta Biomaterialia 108 (2020) 97–110, and BioMed Eng OnLine (2017) 16:40.
We would like to admit that our lack of information about PCL in the manuscript was intentional, as it is a polymer that is quite well known and used and in the field of biomedical engineering and nanobiocomposite engineering. At the express request of the reviewer, we introduced a paragraph discussing the characteristics of this polymer. Unfortunately, due to the five reviewers of our paper (none of the reviewers rejected the manuscript but each of them saw the introduction differently), the introduction is a significant part of the manuscript and does not comply with the rules of the art of writing scientific articles. Nevertheless, we thank you for the suggested literature that we included in the manuscript
Q4. In materials Methods section authors should clearly provide the details of PCL and MMT amount (in wt%) in electrospinning process. Page 3, line 140 authors calculate 2 wt.% amount of filler, which is also mentioned in the Abstract (page 1 line 13). What is the amount of PCL ? Pls also write the details of gentamicin amount. Please mentioned it.
Thank you for your comment. All data were included in the manuscript; both the amount of nanofillers (which was 2% by weight in each case) and the concentration of PCL (15% by weight). In turn, the effective amount of gentamicin was developed based on the methodology described in our earlier work [1-2]. The known amount of gentamicin allows for further elaboration of the release kinetics of gentamicin from the tested materials. Although the method of its introduction is not the subject of this work, such information could affect the readability of the manuscript.
[1] A. Rapacz-Kmita, M.. Bućko, E. Stodolak-Zych, M. Mikołajczyk, P. Dudek, M. Trybus, Characterisation, in vitro release study, and antibacterial activity of montmorillonite-gentamicin complex material, Mater. Sci. Eng. C. Mater. Biol. Appl. 2017, 70 (Pt 1), 471–478. https://doi.org/10.1016/J.MSEC.2016.09.031
[2] A.Rapacz-Kmita et al. Magnesium aluminium silicate–gentamicin complex for drug
delivery systems. Preparation, physicochemical characterization and release profiles of the drug J Therm Anal Calorim (2017) 127:871–880 doi 10.1007/s10973-016-5918-4
Q5. Figure 1, Why gentamicin does not exhibit any peak in XRD analysis? What does it mean?
As already mentioned in the manuscript (line 227-229), gentamicin sulfate is an amorphous substance, therefore in the XRD plots a curve without reflections was registered, which is typical for amorphous substances.
Q6. PCL is a hydrophobic material. Figure 7, why addition of MMT, G, and MMTG in PCL reduce the water contact angle? Authors should provide bar graph.
The statement above is supported by the literature. Sołtysiak et al. show that even cast PCL films into which MMT particles were introduced gain in roughness - the presence of MMT particle influences the ease of polymer spherulites formation and MMT itself can act as a factor initiating nucleation of polymer chain. In the discussed case of fibrous material, we can observe the effect of increase in the fiber diameter due to the presence of MMT filler in its volume (Figure 4 in the manuscript). An increase in the fibre diameter and a wider fibre size distribution is an effect of 'smoothing' of the nonwoven surface, and thus a better and faster penetration of the surface by water droplets (easier dispersion), which results in a smaller value of the wetting angle. Further decrease in contact angle can be caused by the presence of gentamicin not only between montmorillonite layers but also by the presence of particles of genramycin sulfate salt on the surface of powder added to electrospinning. This assumption is confirmed by almost identical value of contact angle for PLA_G and PLA_MMTG materials.
[1] E. Sołtysiak, M. Błażewicz, influence of nanoparticles on physical and chemical surface material properties, Eng Biomat, 92, (2010), 30-3
Q7. Figure 8, authors should provide tensile stress vs tensile strain curve as explain in (Fig.4 of ref. ACS Appl. Mater. Interfaces 2015, 7, 8088−8098). Why the tensile strain for nanocomposite nanofibers is higher than the pure PCL nanofibers? The data for PCL-MMTG and PCL-MMT showing fluctuations, why? It would be good if authors can provide smooth curve for these two materials.
Thank you for your attention. In the suggested manuscript we are dealing with a material that, although based on PCL fibers, is additionally coated with a layer obtained by sol-gel method based on silica. This material, called by the authors as hybrid, has a different nature than the fibers studied by us. In the work of Rajendra K. Singh, a material is studied which, due to post-spinning modification, has the character of a ceramic material; it is brittle, so mechanical data are collected in a tensil strengh-stress system. The authors of this paper write that: the stress versus strain response of the scaffolds MS@PCL was strongly affected by the silica coating on the surface. They further explain that: The MS layered on the surface was highly effective in increasing the mechanical strength and the bulk elastic modulus of the polymeric matrices while compensating for the elongation behavior of PCL.
Here we are dealing with a material that is created at the manufacturing stage (prespinning modification). A 2%wt/wt nanofiller is introduced in order to induce an appropriate interaction between the nanofiller and the polymer matrix (polymer chain). Many papers on nanocomposites use mechanical tests to confirm the high dispersion of the particle in the polymer matrix; they are performed in the force-strain regime to determine the elongation of the material. The fibers are arranged randomly, hence the fluctuations in the tensil strength-elongation curves. Of course, mathematical treatment of the data allows the curves to be smoothed, but the data shown in this way is more true to character - it reflects the nature of the material. Our material is a highly elastic material and the effect of the particle on the polymer can only be determined in the system used. This is the reason why we observe an increase in the breaking strength values in nanocomposite systems: the polymer chain stressed during the test approaches the nanoparticle and there is then a chance to induce secondary interactions such as dispersion forces, wan der Waals or London interactions.
Q8. The release of gentamicin showing burst release at early time point and exhibits long-term release up to 12 days why? Please explain what mechanism involve in the release process?
Thank you for your question. Our previous work on the release of gentamicin sulfate from a carrier such as MMT was preceded by a literature review that dictated our research methodology to show the kinetics of salt release. Based on the data available in the drug database [1], which describes the maximum daily dose of the drug (10 ÷ 12 μg/ml), the obtained gentamicin concentration in the PBS solution was recalculated to the maximum value of the drug (10 μg/ml). The kinetic of drug release was evaluated based on the zero-order, first-order, Higuchi and Korsmeyer–Peppas models [2].
[1] Clinical Guideline for Gentamicin Prescribing and Therapeutic Drug Monitoring,
Royal Cornwall Hospitals NHS, 2014.
[2] A.Rapacz-Kmita et al. Magnesium aluminium silicate–gentamicin complex for drug delivery systems. Preparation, physicochemical characterization and release profiles of the drug J Therm Anal Calorim (2017) 127:871–880 doi 10.1007/s10973-016-5918-4

Reviewer 5 Report
Authors evaluated the effect of montmorillonite addition on the properties of electrospun polycaprolactone fibers. The component they used is interesting and novel. I suggest major revisions before being published.
- Figure 11, cumulative % should be within 100%, they showed in the wrong format. Please correct.
- Many typical errors are there. The grammar correction is not needed to show. Please remove in revised manuscript.
- Line 120, the spelling error about PCL.
- Why authors did not supply XRD of PCL? need to elaborate with relevant ref such as
Carbohydrate Polymers, Volume 250, 15 December 2020, 116880, Scientific Reports volume 9, Article number: 2943 (2019)
- XRD range should show in a broader range.
- I saw they present data of many digits even decimal fractions. I think this is not necessary. The accuracy of data can not be obtained by instruments.
- Figure 3/4/5 can be merged.
- The figure size should be consistent throughout MS.
- Similarly, Figure 7/8/9 can be merged.
- I am surprised why the PCL mat also shows antibacterial properties in contrast to many reports. I think this is the mistake authors made.
- Figure 8, x-axis label was a mistake. Please correct. The discussion about Figure 8 and Figure 9 was poor. It should be elaborated. These references from comment number 4 can be helpful.
Author Response
Dear Reviewer,
Thank you for careful reading our manuscript entitled: Effects of montmorillonite and gentamicin addition on the properties of electrospun polycaprolactone fibers (Materials-1361097). We hope that the answers below will provide a better explanation of our manuscript.
Q1. Figure 11, cumulative % should be within 100%, they showed in the wrong format. Please correct.
As requested by the reviewer, the figure has been revised and included in the manuscript
Q2. Many typical errors are there. The grammar correction is not needed to show. Please remove in revised manuscript. Line 120, the spelling error about PCL.
The manuscript has been submitted for grammatical proofreading by a native speaker with a technical background. We hope that the current language will facilitate the reception of the manuscript. Corrected and highlighted in yellow in the text.
Q3. Why authors did not supply XRD of PCL? XRD range should show in a broader range. Need to elaborate with relevant ref such as Carbohydrate Polymers, Volume 250, 15 December 2020, 116880, Scientific Reports volume 9, Article number: 2943 (2019)
In our manuscript, we studied a fibrous nanocomposite material in which the amount of nanofiller (MMT, MMTG or G) does not exceed 2%wt/wt. Additionally, performing XRD on a fibrous membrane whose thickness is only 180-250um is difficult to realize. The information we wanted to obtain, i.e. the role of the filler in the matrix and the possibility of changing the structure of the material from amorphous to crystalline, is possible and easier to obtain in thermal DSC tests. There the weight of the material is about 3.5mg which gives more reliable results. In the suggested paper, the authors of the paper analyzed composite materials: Al-doped ZnO/cellulose composites where the proportion of additive to the polymer matrix was 10, 20 and 30%wt/wt, respectively. Unfortunately, it is not possible to relate the studies performed there to our case.
Q4. I saw they present data of many digits even decimal fractions. I think this is not necessary. The accuracy of data cannot be obtained by instruments.
We completely agree with the reviewer's comment, so the excessive precision in the number of digits has been removed.
Q5. Figure 3/4/5 can be merged. Similarly, Figure 7/8/9 can be merged. The figure size should be consistent throughout MS.
As requested by the reviewer, the figures have been merged and their size are consistent throughout MS
Q6. I am surprised why the PCL mat also shows antibacterial properties in contrast to many reports. I think this is the mistake authors made.
Indeed, there is no information in the literature on the bactericidal activity of PCL alone, although the results we obtained indicated a zone of inhibition - hence the above statement in the text of the manuscript. We repeated microbiological tests according to the protocol of the American Society for Microbiology on the discs with PCL alone [1]. The results obtained are presented below (Figure 1). The minimal inhibition zone and the precise encirclement is a result of overlapping of the nonwoven disc and its breaking into the agar, hence the different color of the culture around the tested material.
Fig. 1. Microbiological tests on PCL discs; the visible "inhibition zone" is the result of gentle pressing of the disc into the agar on which bacteria were seeded.
Thanks to the reviewer's attention we performed comparative tests with the dynamic condition dedicated to textiles [1]. Their last stage is the study on the series of dilutions in which it is possible to observe the behavior of bacterial colonies due to the nature of the interaction between the material and the inoculum.
The disparities observed between the well diffusion and dilution data could be attributed to various factors that include, but are not limited to, diffusion efficiency of the test compound, sample polarity and concentration. Due to these limitations, dilution assays are often considered more reliable and reproducible compared to the solid agar diffusion assays
We have corrected an incorrect sentence in the manuscript
[1] J. Hudzicki, Kirby-Bauer Disk Diffusion Susceptibility Test Protocol, American Society for Microbiology © 2016
[2] JIS L 1902: Testing Antibacterial Activity and Efficacy on Textile Products, 2002
Q7. Figure 8, x-axis label was a mistake. Please correct. The discussion about Figure 8 and Figure 9 was poor. It should be elaborated. These references from comment number 4 can be helpful.
In the literature, there are many reports based only on the intercalation of nanoclay alone. In many cases this solution works well, as it allows the controlled release of catalysts that initiate chemical reactions, salts that complex contaminants or drug carriers [1-3]. Unfortunately, in many cases the nanoparticle form itself can be an unacceptable form due to its size and not fully predictable behavior when such a particle enters the microorganism. The process of release of such substances can be slowed down or even further controlled by introducing intercalated nanoclayers into the polymer matrix. In this case, however, a suitable carrier form of such nanoclay is required, enabling the action of active substances. Here, porous microstructures, including fibrous membranes, are useful, especially since studies show that the form of electrospun fibers is suitable for controlled release of drugs and other active substances and the form of membrane facilitates such applications [4-6].
The layered aluminosilicate modified with sulfate in aqueous solution such as phosphate buffer is well penetrated by water and ionic components of the buffer, thus removing sulfate from the galleries between the MMT layers. This results in an increase of the sulfate concentration in solution (Figure 9). On the other hand, the incorporation of the intercalated filler into the polymer matrix protects it from strong water penetration; the polymer layer protects the active compound in the aluminosilicate gallery. Consequently, there is a slower release of sulfate for the ma-teria PCL_MMTG which is evident from the lower concentration observed after 6 yaks and 216h of observation. Gentamicin sulfate is released more rapidly in the system where it is not directly entrained with the fiber-forming polymer layer (PCL_G). The wettability of MMT_G and PCL_MMTG membranes is not insignificant.
[1] P.B. Messersmith, E.P. Giannelis, Synthesis and barrier properties of poly(ε-caprolactone)-layered silicate nanocomposites, J. Polym. Sci. Part A Polym. Chem. 1995, 33 (7), 1047–1057. doi.org/10.1002/POLA.1995.080330707
[2] Nafeesa Khatoon et al., Nanoclay-based drug delivery systems and their therapeutic potentials J. Mater. Chem. B, 2020,8, 7335-7351, doi.org/10.1039/D0TB01031F
[3] Selvakumar Murugesan et al., Copolymer/Clay Nanocomposites for Biomedical Applications, Adv. Funct. Mater. 2020, 30, 1908101, pp 1-28 doi: 10.1002/adfm.201908101
[4] A. Walther, et al., Large-area, lightweight and thick biomimetic composites with superior material properties via fast, economic, and green pathways, Nano Lett. 2010, 10, 2742.
[5] R. F. Fakhrullin, et al., Halloysite clay nanotubes for tissue engineering, Nanomedicine 2016, 11, 2243.
[6] S. Srivastava, et al., Novel shape memory behaviour in IPDI based polyurethanes: Influence of nanoparticle, Polymer 2017, 110, 95.

Round 2
Reviewer 1 Report
N/A
Author Response
Dear Reviewer,
Thank you for your work
Reviewer 2 Report
Some punctuation errors still remain throughout, please check for consistency:
Decimal should include a period rather than a comma: Example: lines 17-18: 12,9 A should be 12.9.
Some references are missing the space subsequent the comma: Example: Lines 103-108 [23,29].
The 'Error reference not found' is still present in line 323, 350.
Author Response
Dear Reviewer,
Thank you for careful reading our manuscript entitled: Effects of montmorillonite and gentamicin addition on the properties of electrospun polycaprolactone fibers (Materials-1361097). We have carefully read all of the reviewer's suggestions and have made the indicated editorial changes to the manuscript. We hope that the manuscript in its present form will be accepted by the reviewer:
- commas replaced with periods,
- missing literature was completed,
- erroneous indications were removed (ie. error ....)
All corrections have been highlighted

Reviewer 3 Report
Accepted in the present form
Author Response
Dear Reviewer,
Thank you for your work
Reviewer 4 Report
Authors have improved the manuscript and also provided satisfactory responses of the comments. Therefore, I recommend for publication.
Author Response
Dear Reviewer,
Thank you for your work